# Rapid dynamics of dorsal raphe serotonin neurons regulate the strength of visual attention

Jonas Lehnert[1,2,3], Kuwook Cha[1], Julia Forestell[1], Kerry Yang [1], Xinyue Ma[1], Sinan Shariff[1], Jonathan P. Britt [4], Anmar Khadra [1,2,3], Erik P. Cook [1,2,3] ✉ & Arjun Krishnaswamy [1,2,3] ✉

Every moment, a spotlight of attention roams our visual field to enhance detection of salient stimuli. While recent work suggests how the brain selects salient relevant stimuli (attentional focus), it is unknown if there are mechanisms that modulate attention to suit fluctuating target salience or internal needs (attentional strength). Here, using optical recordings and perturbations in mice performing a cued detection task, we report that rapid dynamics in both dorsal raphe (DR) activity and cortical serotonin (5HT) release selectively regulate attentional strength. In contrast to these rapid DR-5HT dynamics, we found previously reported reward-associated and slower dynamics were both unrelated to attention. Brief optogenetic DR-5HT suppression increased attentional strength and performance, whereas optogenetic excitation reduced attentional strength and performance. A computational model suggests DR-5HT's inverse effects on attention arise from control of cortical suppression and divisive normalization. Collectively, our results demonstrate that the brain independently controls attentional strength and focus and defines rapid DR-5HT neuromodulation as a new regulator of attentional strength.

Visual attention enhances detection of task-relevant visual targets[1–4]. Significant effort has been devoted to how the brain selects target locations and features[1–10] (referred to as attentional focus). However, the same targets could be weakly or strongly attended (attentional strength). Control of attentional strength, independent of attentional focus, could let the brain tailor enhancement of stimuli to meet fluctuating target salience or internal need. However, evidence for this separation is currently unavailable. Moreover, while many studies suggest cortical mechanisms for attentional focus[1–10], less is known about the mechanisms underlying attentional strength.

Neuromodulators have been proposed as candidates for control of visual attention[11–14], and of these, serotonin (5HT) is particularly intriguing. Forebrain 5HT originates from projection neurons in the dorsal raphe (DR) that encode internal states such as reward[15–17] or patience[18–22], and track uncertainty[23–26] in various behavioral paradigms. DR neurons innervate many cortical areas, including visual, parietal, and frontal areas[27–31]; such areas are believed to hold distributed representations of attention targets[1–10]. Recent optogenetic studies reveal that DR-5HT modulates both baseline and visually driven responses in V1 neurons[32,33], which echoes prior 5HT iontophoresis work in non-human primates[34–36]. Thus, 5HT reflects goal-related internal states, is released in both visual and frontal areas, and modulates visual neuron excitability. Yet, whether 5HT is involved in visual attention is unknown.

To address this question, we optically monitored and manipulated DR-5HT signals while mice performed a cued visual detection

[1]Department of Physiology, McGill University, Montréal, QC, Canada. [2]Quantitative Life Sciences, McGill University, Montréal, QC, Canada. [3]Center of Applied Mathematics in Bioscience and Medicine, McGill University, Montréal, QC, Canada. [4]Department of Psychology, McGill University, Montréal, QC, Canada. ✉e-mail: erik.cook@mcgill.ca; arjun.krishnaswamy@mcgill.ca

task and analyzed their responses with a behavioral model that independently measured attentional strength and focus. Simultaneous fiber photometry revealed that rapid fluctuations in DR neural activity predicted attention to cued stimuli; brief DR suppression predicted strong attention and performance, whereas brief excitation predicted weak attention and performance. 5HT release dynamics in the visual cortex mimicked DR dynamics and also predicted attentional strength. Rapid DR-5HT dynamics had no impact on attentional focus, nor did it affect states of arousal, impulsiveness, or engagement. Optogenetic excitation of DR neurons deteriorated attentional strength and performance, whereas optogenetic suppression increased attentional strength and performance. Neither optogenetic perturbation affected attentional focus, nor did they affect other internal states. A computational model suggests DR-5HT's inverse effects on attention arise from control of cortical suppression and divisive normalization. These results demonstrate the separability of attentional strength and focus and define DR-5HT as a new regulator of attentional strength.

## Results

### Behavioral measurements of attention from a noisy cued-detection task

We used a cued detection task that controls contextual signals related to attention and measures other internal states that influence perception and performance[37]. Briefly, water-restricted mice were head-fixed atop a platform facing a pair of angled screens and trained to search for 3 white vertical bars that emerged from dynamic checkerboard noise (Fig. 1a). We varied trial difficulty by varying the amount of noise combined with the 3-bars (Fig. 1b, *coherence*). Trials began with a tone and static checkerboard on one screen which cued the eventual

location of the 3-bar grating (Fig. 1a). Mice licked a spout to start trials (*start lick*), which caused the cue to fade while zero-coherence dynamic checkerboard noise was presented on both screens for a random delay period (3–12 s) until the grating appeared (Fig. 1a, *delay*). A lick within the reaction time (RT) window after grating appearance (*hits*) led to a liquid reward. Gratings switched sides after ~35 trials and had a flat probability of appearance in time. Since gratings were noisy and appeared unpredictably, our task forced mice to continually evaluate checkerboards for evidence of the 3-bars. Trained mice detected weak gratings reliably (d-prime ≥ 1, coherence = 0.3, Fig. 1c).

Licks to pure noise were false alarms (FAs). FAs occurring within the RT window after zero-coherence grating appearance were unrewarded and used to compute psychometric measures (Fig. 1c). FAs during the delay period caused trials to restart until the delay period was lick-free (Fig. 1a, *restart*). Most mice FA licked to noise, and delay periods were seldom without FAs. We previously showed that many FAs occur because mice react to grating-like features that emerge randomly from the noise. By reverse correlating FAs to the noisy checkerboards, we saw that mice attended to the cued side for increases in verticality and local contrast[37]. Here, we simplified this prior approach to better quantify visual attention.

Briefly, we convolved checkerboards with a vertical Gabor filter to extract vertical energy and a 2D Gaussian filter to extract local contrast energy (Fig. 1d). Next, we binned both energy maps in a 3 x 5 grid (bin = ~11°) encompassing the entire checkerboard. Finally, a logistic regression model weighed energies on cued and uncued screens to predict all FA licks. Fitting this model to each mouse provided two sets of weights that captured: (1) behavioral gains for cued and uncued screens ($\alpha_{Cued}$ and $\alpha_{Uncued}$); and (2) sensitivity to vertical ($\beta_V$) and

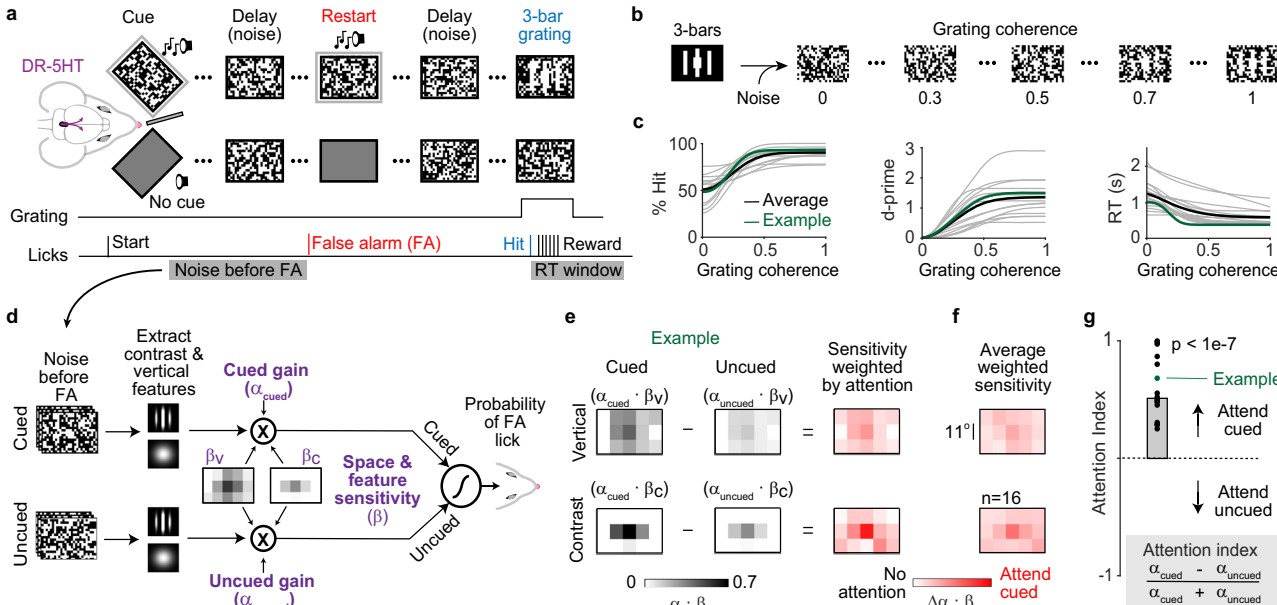

**Fig. 1 | Task to measure attentional focus and gain. a** Cartoon of a head-fixed mouse facing angled visual displays to detect the 3-bar grating. Trials began with a static checkerboard and audiovisual cue that indicated the eventual location of the 3-bar grating. The cue quickly faded after mice licked (Start) and was replaced by dynamic checkerboard noise (30 Hz) presented for a randomly chosen delay. A lick within a 1.5 s reaction time (RT) window following the 3-bar grating (Hit) resulted in a reward. A lick during the delay period (False alarm, FA) restarted the cue and trial. FA restarts recurred until the delay period was lick free. **b** Example gratings produced by combining 3-bars with different amounts of checkerboard noise (52 x 34° of visual angle). All 3-bar checkers are white when coherence = 1, they are all noise when coherence is 0. **c** Psychometric curves from trained mice relating %hits, d-

prime, and RT to stimulus coherence. (Gray lines = individuals, black = mean, green = example mouse, ~3000 trials/mouse; n = 16). **d** Individual checkerboard frames are convolved with vertical Gabor or 2D Gaussian filters to obtain vertical and contrast energy, binned across 15 locations, and fit by a logistic regression model that estimates sensitivity to vertical and contrast energy ($\beta_V$ and $\beta_C$) and behavioral gain ($\alpha_{Cued}$ and $\alpha_{Uncued}$) and predicts the probability of a FA lick. **e** Model results for an example mouse presented as sensitivities to vertical and contrast energy weighed by the difference in behavioral gain (attention). **f** Averaged weighted sensitivity maps show enhanced attention to the center of the cued screen before FAs. Color scale: 0 – 1. **g** Population attention indices show increased attention to the cued side (one sample t-test; n = 13 mice, p = 2.1e-7).

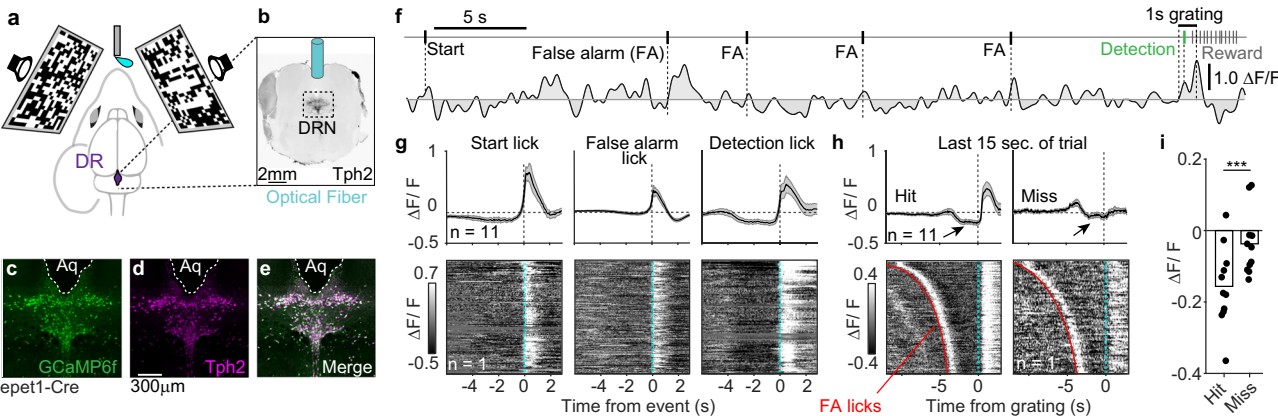

**Fig. 2 | DR neural signals encode trial events and outcomes. a** Cartoon shows fiber photometry from DR while mice detect 3-bar gratings. **b** Coronal brain section immunostained with antibodies against tryptophan hydroxylase 2 (Tph2, black); optical fiber position indicated. Representative **c**oronal DR section from an epet1-Cre x GCaMP6f mouse immunostained for GCaMP6f (**c**), Tph2 (**d**), and a merge (**e**) from the fiber photometry cohort (histology was performed in $n = 11$ mice with similar results across animals). **f** Example photometry trace from a single trial with start, false alarm, detection, and reward licks indicated. **g** Mean GCaMP6f signal (top) and an example raster of raw recordings (bottom) from DR-5HT neurons. Traces are aligned to the indicated trial events ($n = 11$ mice; shading is mean ± SEM). **h** Mean photometry signal and example response raster aligned to grating onset for the last 15 sec of hit and miss trials. DR is suppressed before grating on hit trials (arrows). Rasters are sorted based on the length of the delay period before the grating onset. FA licks prior to delay period indicated in red. **i** Mean GCaMP signal 1.5 sec before grating onset for hits and misses. ($n = 11$ mice; *** $P < 0.001$, paired t-test, $p = 6.6e\text{-}5$).

contrast energy ($\beta_C$) across both screens. We defined an attention index computed from behavioral gains to measure attentional strength, whereas attentional focus was characterized by sensitivity to verticality and contrast (maps of $\beta_C$ and $\beta_V$).

This behavioral model had significant predictive power when fit to each mouse (Supplementary Fig. 1a). Sensitivity maps revealed that mice focused on verticality ($\beta_V$) and local contrast ($\beta_C$) at central screen locations (Fig. 1e, f and Supplementary Fig. 1b), consistent with our earlier study[37]. Moreover, our attentional index quantified that mice weighed cued stimulus energies ~60% more than uncued energies (Fig. 1g). Taken together, these results show that mice attended to bar-like features at the center of the cued screen to perform our task.

### DR neural activity is linked to behavior
Next, we fiber-implanted mice that expressed the genetically encoded calcium indicator (GCaMP6f) within DR-5HT neurons (Fig. 2a–f and Supplementary Fig. 1c–g). We grouped datasets from two DR-5HT Cre lines (epet1-Cre and Cdh13-CreER), because they similarly marked DR-5HT neurons[27–29,31] (Supplementary Fig. 2a–c) and had similar photometry signals (Supplementary Fig. 2d–g).

DR-5HT neural activity fluctuated rapidly across trial length (Supplementary Fig. 1h) and reacted to behaviorally important lick events (Fig. 2f, g). Hits were associated with positive transients after grating presentation, whereas misses were not (Fig. 2h), consistent with prior work on DR activity with reward[15,17]. However, we noticed that average DR-5HT activity just before grating onset (at the very end of the delay period) was lower in hit trials than in miss trials (Fig. 2h, i, *arrows*).

This pre-grating DR-5HT signal differs from prior work on task-related DR signals in a few ways. First, it is an order of magnitude faster than previous reports, which show minute-long, slow DR dynamics[16]. Second, we saw no trial-level correlation between this pre-grating signal and the post-grating response for all hit trials (Supplementary Fig. 2h). Third, we saw no correlation between a hit trial's post-grating response and the next trial's pre-grating signal (Supplementary Fig. 2i). These data indicate that rapid DR-5HT dynamics just before grating onset are separate from post-grating transients associated with rewards. These results also reveal that rapid pre-grating suppression in DR activity precedes hits, whereas pre-grating DR activity in miss trials was more elevated in comparison. Given these results, we next asked if rapid DR-5HT dynamics were linked to task performance.

### DR neural activity predicts behavioral performance and attention
We began by measuring task performance for trials where DR-5HT activity was low or high before grating presentation (Fig. 3a). Psychometric measures (% hits, d-prime, threshold) showed that mice detected low-coherence gratings better when DR was low as compared to when it was high (Fig. 3b–d); no changes were seen in reaction time, response bias, lapse rate, or criterion (Fig. 3e). These results show that low DR activity predicts higher sensitivity to gratings (d-prime, hits, threshold) rather than changes to impulsivity (response bias), engagement (lapse rate), or strategy (criterion). The performance difference between trials sorted by pre-grating DR-5HT high and low faded away at high coherences; this is likely why pre- and post-grating $\Delta F/F$ over all hit trials was uncorrelated (Supplementary Fig. 2h). Finally, these links to performance were uniquely associated with fast dynamics, since the same analysis for slow DR-5HT dynamics (Supplementary Fig. 3a–e) and fast-slow interactions (e.g., fast suppression on a slow rise versus on a slow fall, Supplementary Fig. 3f–h) showed no correlation with task performance.

We hypothesized that the enhanced detection of weak coherent gratings during low DR activity (Fig. 3b–d) reflected the well-known ability of attention to enhance weak stimuli[1–4]. If true, then our behavioral measures of attention should be correlated with DR activity. Specifically, low DR activity could improve attentional focus (maps of $\beta_C$ and $\beta_V$), could increase attentional strength (attention index), or could improve both attentional strength and focus. To test this idea, we divided FA licks based on whether they were preceded by low or high DR activity (Fig. 3f) and then measured attentional strength and focus.

Sensitivity maps weighted by attention revealed emphasis on verticality and local contrast at the center of the cued side when the DR was low; this emphasis was qualitatively weaker when the DR was high (Fig. 3g, h). Sensitivity to vertical energy and local contrast remained at the center of the screen in both DR high and low (Fig. 3h and Supplementary Fig. 4a, b). In contrast, attentional indices from DR low were significantly stronger than those computed from DR high (Fig. 3i). These results indicate that rapid DR low dynamics are correlated with increases in attentional strength rather than changes in how mice allocated vertical/contrast sensitivity across the screen (attentional focus).

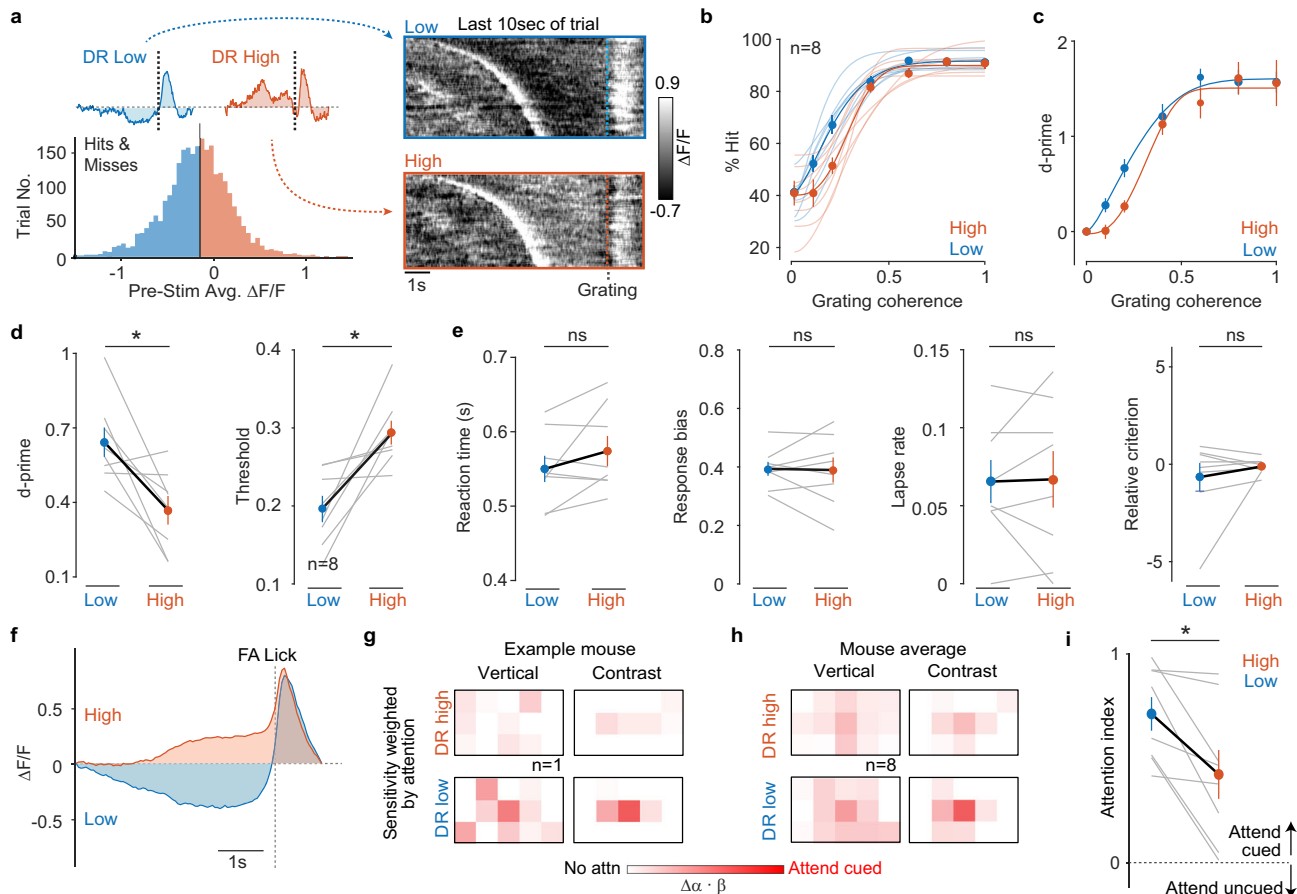

**Fig. 3 | DR neural activity predicts attention and detection of the 3-bar grating.**
**a** Histogram of mean DR signal preceding grating presentation divided into low-
and high-activity groups for one mouse. Insets show mean response activity from
each group. Raw single-trial response rasters for low and high groups aligned to
grating onset also shown. Psychometric curves of % hits (**b**) and d-prime (**c**) versus
stimulus coherence for DR high and low. Mice perform with higher sensitivity in DR
low trials than in DR high trials. Light lines are 8 individual mice; thick lines are the
mean. **d** Mean d-prime at 0.2 coherence and psychometric threshold for DR high
and low. Threshold is significantly reduced and d-prime significantly elevated when

DR is low. ($p$ = d': 0.014, threshold: 0.016) **e** Reaction time (RT) at 0.2 coherence,
response bias, lapse rate, and criterion for DR high and low. **f** Average DR high and
low signals before a false alarm (FA). Example (**g**) and average (**h**) maps of sensi-
tivity weighted by attention for DR high and low ($n$ = 8). DR low enhances attention
to cued side. Example mouse color scales: DR high = 0–0.2; DR low = 0–1. Average
mouse color scales: 0–1 for DR high and low. **i** Mean attention index for low and
high DR activity ($n$ = 8 mice; $p$ < 0.0078). Attention index is significantly increased
when DR activity is low. * $P$ < 0.05, Wilcoxon signed-rank test. Error bars in **b**–**e**, **i** are
mean ± SEM.

We next analyzed behavioral attention measures just after our cue
changed sides. Previous studies revealed that subjects often show
poorer performance at block switches in cued tasks because it takes
time for them to reorient attention to the new cued location or
feature[38–42]. To address this idea, we compared attentional strength and
focus between two pairs of FAs: (1) FAs preceded by low DR-5HT
occurring early in the block (after the cue switched sides) and all FAs; (2)
FAs preceded by high DR-5HT occurring early in the block and all FAs.
This comparison revealed weaker attention indices early in the block for
both DR-5HT high and low FAs as compared to all other FAs (Supple-
mentary Fig. 4c), consistent with prior work[38,39]. However, early-block
attention indices were higher when DR was low as compared to when it
was high (Supplementary Fig. 4c). Importantly, all these changes in DR-
5HT activity and attentional strength arose without effects on sensitivity
to vertical and contrast energy (Supplementary Fig. 4d, e). Thus, our
results so far indicate that rapid DR-5HT dynamics selectively predict
attentional strength rather than attentional focus.

## Optogenetic manipulation of DR neurons changes behavioral performance and attention
The inverse correlation between attentional strength and rapid DR-
5HT dynamics strongly suggested that DR-5HT regulates attention. To

test this hypothesis, we selectively expressed excitatory or inhibitory
optogenetic actuators in DR-5HT neurons (Fig. 4a, b) and asked whe-
ther suppressing or elevating their activity could alter attention and
detection behavior (Fig. 4c). Briefly, mice expressing light-activated
excitatory (ChR2) or inhibitory (Jaws) opsins in DR-5HT neurons were
implanted with optical fibers and fully trained on the task before
optogenetic manipulations began. Three seconds of light from an LED
was used to drive either Jaws or ChR2 beginning two seconds before
each grating presentation (Fig. 4d). If the mice FA licked during the
stimulation window, the LED was stopped, the cue was shown, and the
trial restarted (Fig. 4d, *Restart*). Since FA licks could occur prior to LED
onset or in the two seconds between the onset of the LED and the
stimulus grating, we could compare attention indices from FAs that
were directly preceded by stimulation (LED on), with FAs in optoge-
netic sessions that were not preceded by stimulation (LED off), as well
as FAs in non-optogenetic sessions (Ctrl).

Exciting DR-5HT neurons significantly increased psychometric
threshold and reduced d-prime for 3-bar gratings with no effect on
reaction time (Fig. 4e, f and Supplementary Fig. 5a). These effects arose
without significant change in response bias, lapse rate, and criterion
(Fig. 4g and Supplementary Fig. 5c), indicating that DR excitation
reduces sensitivity to gratings rather than affecting impulsiveness,

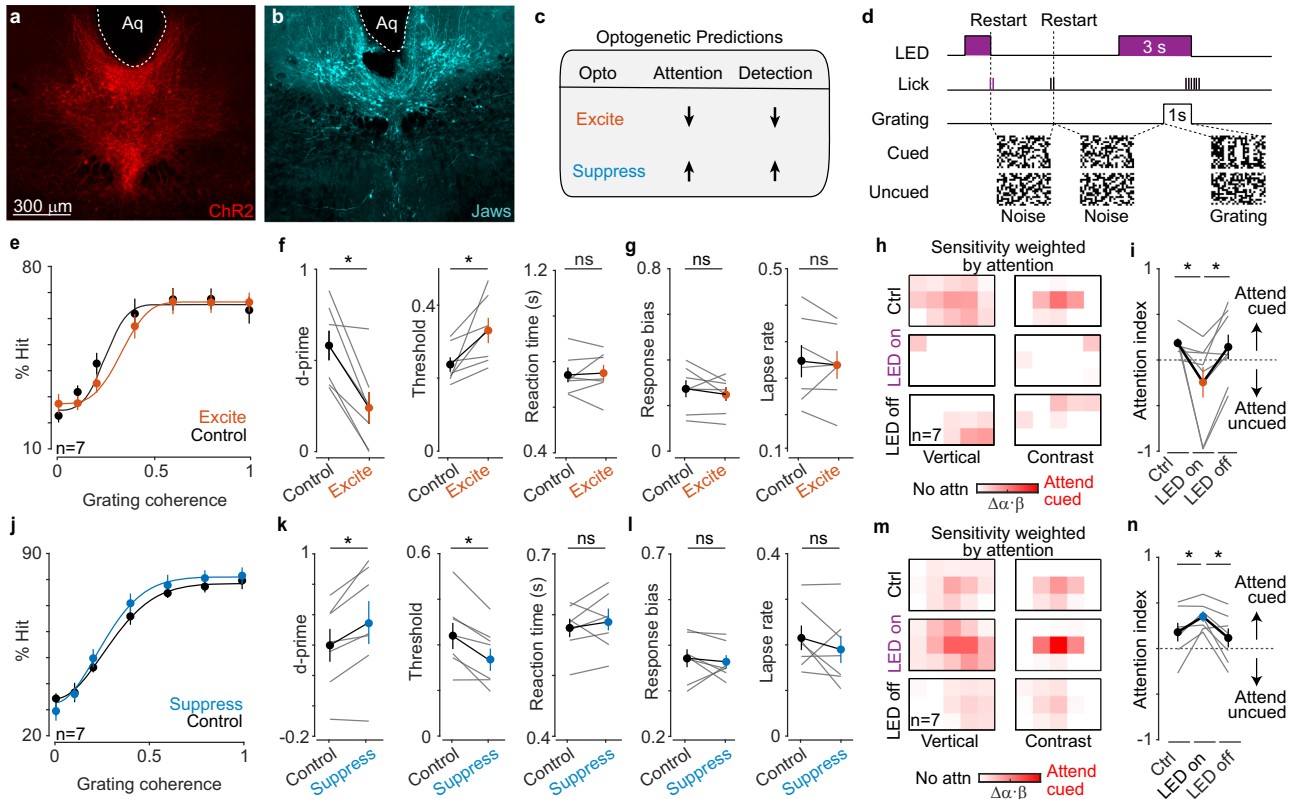

**Fig. 4 | DR regulates visual attention.** Representative cross-sections of DR showing neurons expressing channelrhodopsin (**a**, ChR2) and Jaws (**b**) from optogenetic cohorts (histology was performed in $n = 14$ mice with similar results across animals). **c** Optogenetic excitation (ChR2) and inhibition (Jaws) should suppress or enhance attention and detection, respectively. **d** A train of LED pulses began 2 s before the grating and stopped when the grating disappeared. False alarms (FAs) interrupting LED stimulation caused the trial to restart. **e** % Hits versus stimulus coherence for control and ChR2-DR excitation (Excite). **f** d-prime, psychometric threshold, and reaction time computed from analyses in (**e**). Activating DR deteriorates performance ($p = d'$: 0.016, thresh: 0.016). **g** Response bias and lapse rate computed from the data shown in (**e**). **h** Average weighted sensitivity maps computed from FAs prior to optogenetic stimulation (Ctrl), FAs preceded by stimulation (LED On), and FAs not preceded by stimulation (LED Off). Attention to the cued

screen dropped during DR stimulation. Color Bar: 0–0.4 **i** Average attentional index for conditions shown in h. DR stimulation reduces attentional gain ($p = $ ctrl vs. LED On: 0.016, ctrl vs. LED Off: 0.031). The LED On population is not statistically different than zero (one sample t-test). **j** % Hits versus stimulus coherence for control and Jaws-DR inhibition (Suppress). **k** d-prime, psychometric threshold, and reaction time computed from analyses in (**j**). Suppressing DR enhances performance ($p = d'$: 0.031, thresh: 0.016). **l** Response bias and lapse rate computed from the data shown in (**j**). **m** Average weighted sensitivity maps computed from Ctrl, LED On, and LED Off FAs ($n = 7$ mice, Color Bar: 0–0.4). **n** Average attentional index for conditions shown in (**m**). DR suppression increases attentional gain. *$P < 0.05$, Wilcoxon signed-rank test ($p = $ ctrl vs. LED On: 0.031, ctrl vs. LED Off: 0.031). d-prime and reaction time computed at 0.2 coherence. Error bars in all panels are mean ± SEM. ($n = 7$ mice for **e**–**i** and $n = 7$ for **j**–**n**).

engagement, or strategy. We saw weaker weighted sensitivity maps and significantly reduced attentional indices when the DR was excited as compared to both unstimulated control conditions (Fig.4h, i, *LED on vs Ctrl* and *vs LED off*; *LED on* population is not significantly different than zero even though the fitted values for two mice showed uncued bias). Importantly, this optogenetic reduction in attention arose without any change in lick rate (Supplementary Fig. 5d), indicating that DR-5HT excitation deteriorates attention to cued stimuli rather than changing other internal states such as arousal or engagement. Thus, we conclude that elevating DR-5HT activity selectively reduces attentional strength and deteriorates performance.

Optogenetic suppression of DR-5HT neurons had the opposite effect and significantly increased d-prime and decreased psychometric threshold with no effect on reaction time (Fig. 4j, k and Supplementary Fig. 5b). Again, these improvements arose without significant changes to reaction time, response bias, lapse rate, criterion, and overall lick rate (Fig. 4l, Supplementary Fig. 5c, d), indicating that low DR activity increased sensitivity to cued gratings rather than changing impulsivity, engagement, or strategy. We saw stronger weighted sensitivity maps and significantly increased attentional indices when the DR was suppressed as compared to both unstimulated control conditions (Fig. 4m, n, *LED on* vs. *Ctrl* and vs.

*LED off*). No performance or sensitivity changes were seen when we optically stimulated a sham control mouse (Supplementary Fig. 5e, f). Moreover, spatial sensitivity to vertical and contrast energy remained centered during both kinds of optogenetic perturbation (Supplementary Fig. 5g–j). These results indicate that DR-5HT suppression selectively strengthens a mouse's attention to cued grating-like patterns rather than impacting arousal, engagement, or attentional focus.

These optogenetic studies support our photometry observations and indicate that rapid DR activity selectively regulates attentional strength without changing attentional focus or other internal states.

## V1 serotonin release dynamics predict visual attention and behavioral performance

Since prior work reported neural correlates of attention in mouse V1[38–42], we next asked if the DR-5HT attention signal appears in this cortical area. For this experiment, we delivered AAVs bearing the genetically encoded 5HT sensor GRAB5HT[43] into V1 and wide-field imaged fluorescent signals while mice performed our task (Fig. 5a). Tissue sections through V1 showed broad expression of GRAB5HT across cortical laminae (Fig. 5b), and GRAB5HT showed similar task-related signals as in our photometry dataset (Supplementary Fig. 6a).

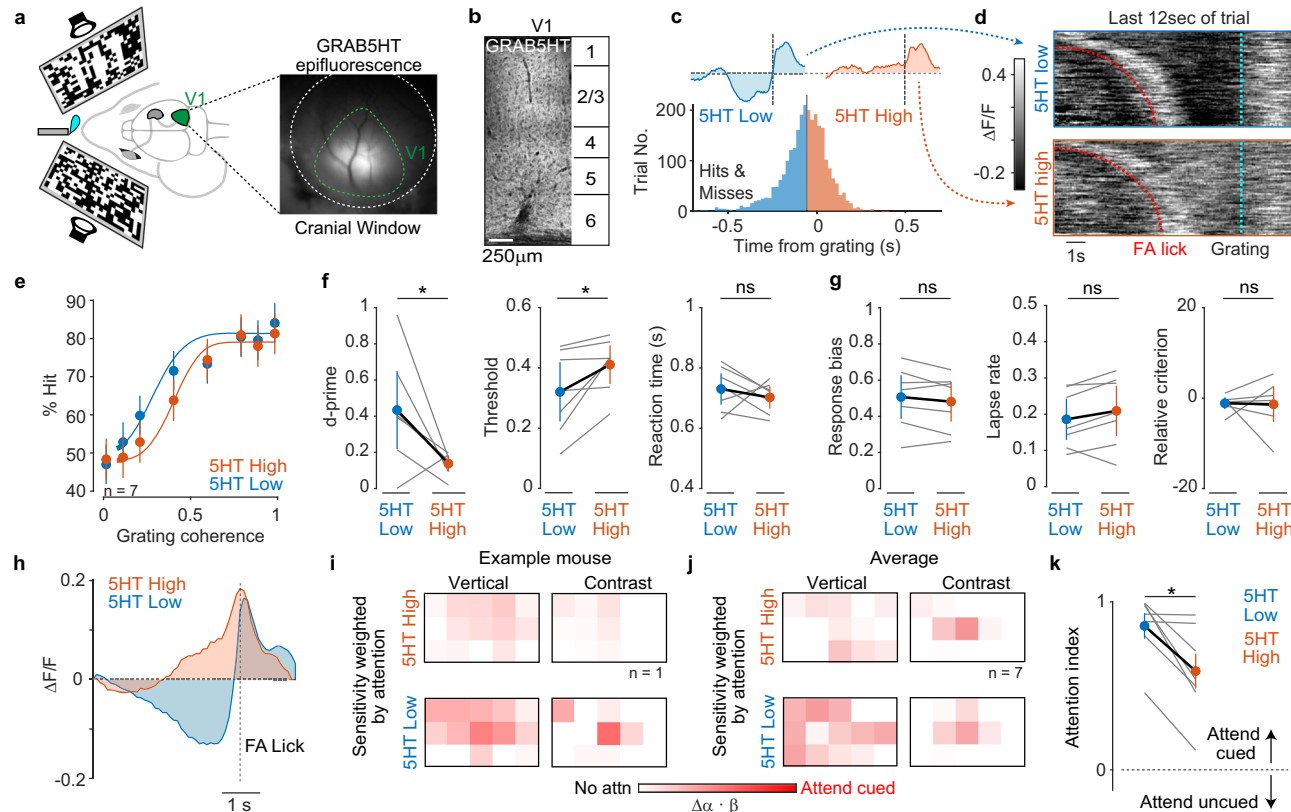

**Fig. 5 | 5HT release in V1 predicts attention and performance.** Cartoon showing expression of GRAB5HT in V1 while mice detect 3-bar gratings (**a**), and an example image through a cranial window; V1 border labeled. **b** V1 tissue section showing GRAB5HT expression throughout cortical depth. **c** Histogram of mean GRAB5HT signal preceding grating presentation divided into low- and high-activity groups for one mouse. Insets show mean response from high and low groups. **d** Raster plots from one mouse showing raw GRAB5HT signals from 5HT low and 5HT high groups. Trials were sorted by the length of the delay period. **e** Psychometric curves from low GRAB5HT (5HT low) and high GRAB5HT (5HT high) groups. **f** Average values for d-prime, psychometric threshold, and reaction time computed from data in (**e**) ($n = 7$, $p = $ d': 0.031, thresh: 0.016). **g** Response bias, lapse rate, and criterion computed from data in (**e**). **h** Average GRAB5HT signal preceding 5HT high and 5HT low FA licks. Example (**i**) and average (**j**) weighted sensitivity maps computed from 5HT high and 5HT low FAs. Attention to the cued side increased when 5HT release in V1 was low. Color scale is 0–1. **k** Attention index from data shown in (**j**) ($n = 7$; $p = 0.016$). Attention index increases during low 5HT release. *$P < 0.05$, Wilcoxon signed-rank test. Error bars in all panels are mean ± SEM.

Importantly, grouping trials by 5HT low and high release in the period just preceding grating appearance (after the long delay) revealed signals resembling our DR photometry data (Fig. 5c, d). Trials with low pre-grating GRAB5HT signals had significantly higher hits, higher d-prime, and lower psychometric thresholds, with no change in reaction time (Fig. 5e, f). These effects arose without significant changes to the response bias, lapse rate, and criterion, suggesting low 5HT release in V1 is associated with better stimulus detection rather than differences in impulsivity, engagement, or strategy (Fig. 5g). Taken together, these results show that low 5HT in V1 predicts better task performance.

We next estimated behavioral attention from FA licks preceded by either high or low GRAB5HT signals (Fig. 5h). We saw qualitatively similar effects on weighted sensitivity maps (Fig. 5i, j) and observed a ~40% larger average attention index during GRAB5HT low as compared to GRAB5HT high (Fig. 5k). Again, these changes in attention index based on the GRAB5HT signal arose without any change in spatial sensitivity maps for vertical and contrast energy (Supplementary Fig. 6b, c). However, the GRAB5HT cohort showed less organized sensitivity maps for vertical energy than those from our photometry and optogenetic cohorts. We previously showed this variability arises because individual mice can choose to monitor contrast energy, vertical energy, or both energies[37]; our GRAB5HT cohort appeared to favor contrast more than vertical energy but this bias was unrelated to 5HT release (Supplementary Fig. 6b). We conclude that 5HT release dynamics in V1 are tightly linked to how strongly a mouse attends but are not related to its spatial sensitivity to stimulus features.

## Discussion

Here, we investigated the role of 5HT in visual attention. By measuring the responses of mice performing a cued visual detection task while optically recording activity from DR-5HT neurons, we found rapid dynamics that selectively predict attentional strength. Optogenetic perturbations showed that rapid DR-5HT dynamics regulated attentional strength, and 5HT sensor measurements showed these dynamics are present in visual cortical areas. Both observed and optogenetically induced DR-5HT dynamics did not affect attentional focus, nor did they affect other states such as arousal or motivation. Collectively, our results show that rapid DR-5HT neuromodulation selectively regulates attentional strength and demonstrate the separability of attentional strength from focus.

### A normalization model of attention with DR-5HT

Our data consistently showed that 5HT inversely regulates attention to the cued screen but did not affect what features were attended. To explore how this mechanism might arise, we adapted the influential normalization model of attention[44] (Fig. 6a). We chose this computational model because normalization is a canonical neural computation[45], a well-accepted feature of visual cortical circuits[44], and because this model accounts for many observed behavioral and neural

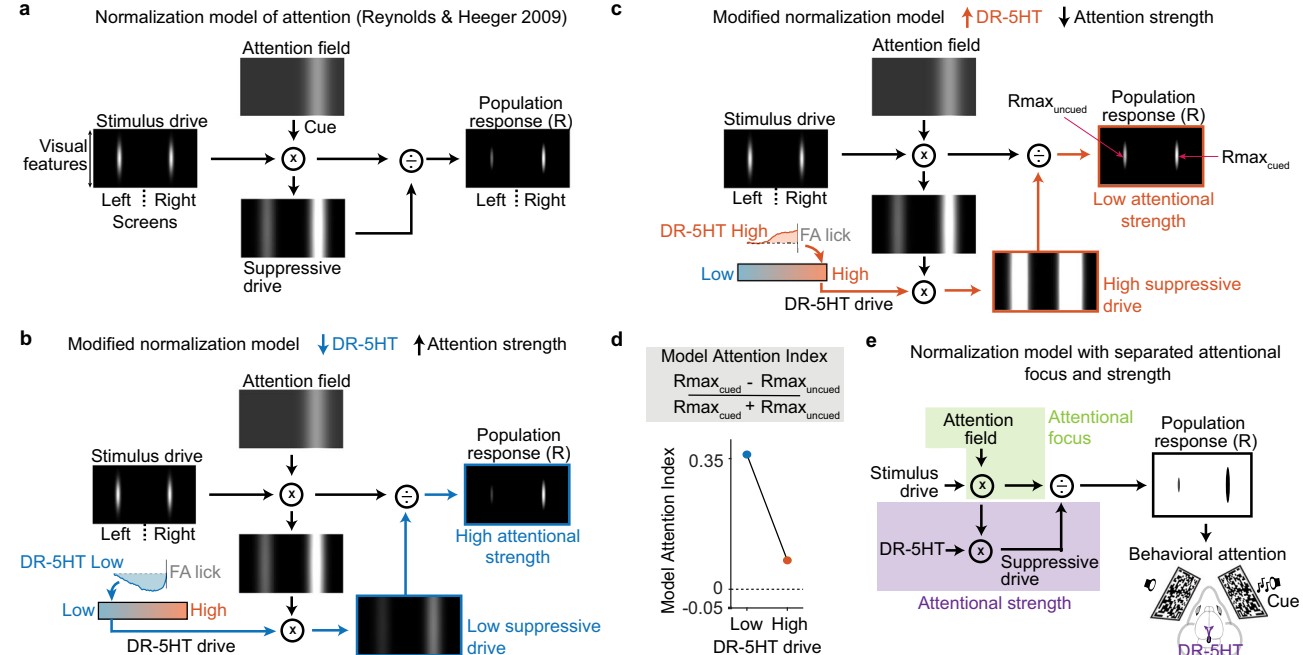

**Fig. 6 | Normalization model of attention and DR-5HT. a** Simulation of the normalization model of attention from Reynolds and Heeger 2009[44]. Stimulus drive represents the neural population response to left and right checkerboard stimuli if there were no attention or normalization. Stimulus drive is depicted with receptive field location along the horizontal and feature preference (orientation, etc) along the vertical. The attention field depicts the effect of cueing the visual location of the right stimulus but is assumed to be non-specific to visual features. The attention field multiplies the stimulus drive, and its product is pooled over space and features to generate the suppressive drive. The panel on the right shows the population response (R) of neurons produced by dividing the product of stimulus drive and attention field by the suppressive drive. **b** Modified normalization model includes a term to scale the suppressive field by DR-5HT (DR-5HT drive). Low DR-5HT

strengthens attention to the cued side. **c** Same as b, but for DR-5HT high which weakens attention to the cued side. Maximum population response to cued and uncued indicated ($Rmax_{cued}$ and $Rmax_{uncued}$). **d** Top: equation to computed model attention index from $Rmax_{cued}$ and $Rmax_{uncued}$; Bottom: Model attention index computed from population responses shown in (**b**) (low DR-5HT drive) and (**c**) (high DR-5HT drive). Higher model attention index when DR-5HT is low versus when it is high mimics our behavioral results showing increased attentional strength. **e** Cartoon of modified normalization model with attentional focus and attentional strength indicated. The attention field focuses the circuit on target features and DR-5HT signals scale suppressive drive to regulate how strongly such targets are amplified in the population responses that underlie behavioral attention.

attentional signatures (eg: contrast gain, response gain, feature-similarity gain, biased competition)[44–48].

Briefly, left and right stimuli drive brain representations that are multiplied by internal attention fields for cued and uncued sides. In this way, cue location selectively boosts one representation without affecting the other (Fig. 6a). Cued and uncued representations are then pooled over features and space to create suppressive drive. Finally, cued and uncued representations are divided by the suppressive drive (normalized) to produce a stronger population response on the cued side (*right stimulus* is cued). Since prior work showed this functional model also explains behavioral effects of attention[44–48], we interpret differences in right versus left model population responses (model attentional strength) as producing our behavioral measure of attentional strength.

We hypothesized that DR-5HT scaled the suppressive drive, a key computation regulating model attention strength. To test this, we multiplied the suppressive drive by a single parameter reflecting DR-5HT and observed the model's predictions (Fig. 6b, c).

Model output aligned exactly with our observations. Model population responses were stronger on the cued side than the uncued side when DR-5HT was low (Fig. 6b). This is because, as the denominator (suppressive drive) shrinks, the attention field's effects on model output grow. Interestingly, cued and uncued population responses were roughly equal when DR-5HT is high (Fig. 6c). This is because as the denominator (suppressive drive) grows, the effect of the attention field weakens. Computing an index from the maximum population response to cued and uncued confirms these observations and shows higher model attention index when DR-5HT is low versus when it is

high (Fig. 6d). Note that DR-5HT changes do not affect the attention field itself (model attentional focus), rather DR-5HT only scales the attention field's effect on the population response (model attentional strength). Thus, our modified model suggests a computational mechanism for separating attentional strength and focus (Fig. 6e).

The modified normalization model captured our general observation that low and high DR-5HT selectively increase or decrease attention to cued stimuli, respectively. It also suggests that this inverse relationship between DR-5HT attentional strength can occur if DR-5HT is part of the suppressive drive. Finally, this model suggests how 5HT release, widely seen as a broad, non-lateralized signal, produces lateralized effects on the cued representation if it scales suppressive drive.

## DR-5HT signals selectively regulate the strength of attention

Rapid fluctuations in DR-5HT selectively produced changes in the attentional index (the measure of attentional strength) but did not affect the location of highest vertical and local contrast sensitivity (attentional focus) and had no effect on states such as motivation. Thus, DR-5HT appeared to change the intensity (strength) of the attentional spotlight but did not change what features it shines upon (focus). This was mimicked by our modified normalization model: DR-5HT modulates suppressive drive without affecting the attention field (Fig. 6b–e). Why would the brain employ 5HT to split attentional control in this way?

One reason could reflect the brain's need to tailor how strongly it attends to suit the fluctuating reliability of visual cues. Thus, a cue engages attentional circuits only if the brain deems it valid enough to

predict future events. Many elegant studies have shown that DR-5HT neurons specialize in predicting reward or punishment from such external cues[23,24,26] and can influence behaviors such as patience[18–22] or persistence[49,50]. Could rapid DR-5HT dynamics encode cue validity and thus modulate attentional strength rather than attentional focus?

Older work on another neuromodulator, acetylcholine (ACh), supports this view[11,51–55]. In these studies, ACh encoded cue validity by tracking expected uncertainty[56]. ACh was high when the outcome (or uncertainty therein) was predicted by a cue and was low when unexpected shifts in context stole predictiveness from the cue[56,57]. Interestingly, recent work indicates that tonic DR-5HT firing is low when expected uncertainty is high[24,25]. Thus, high cue validity could also be encoded as a drop in DR-5HT activity, which would predict strong attention, consistent with what we saw.

However, the audiovisual cue was always valid in our task. So why did attention and DR-5HT fluctuate? One possibility is that grating-like patterns randomly produced by checkerboard noise on the uncued side and/or block switches (Supplementary Fig. 4c–e) stole predictiveness from the cue. Quantifying such expected and unexpected uncertainties over time, as captured by reinforcement learning models[24,56], could shed more light on this issue.

## How could DR-5HT inversely regulate attentional strength?

Our modified normalization model mimicked 5HT's effects on attentional strength if it contributed to the suppressive field (Fig. 6b–e). Prior work suggests a generally suppressive action of 5HT on cortical neurons[32–36]. For example, expression of inhibitory 5HT1A receptor types in pyramidal neurons[58–60] and excitatory 5HT2a receptor types in cortical interneurons[61–65]. Other evidence comes from 5HT iontophoresis studies in non-human primates showing suppressed V1 neuron response gain[34,35] and attenuated V1 population responses to optogenetically-evoked 5HT release[32,33]; interestingly, one of these studies reported 5HT's effects as divisive[32]. Taking these studies together with our model results (Fig. 6), we hypothesize that rapid DR-5HT dynamics multiplicatively scale neural tuning during visual attention. We previously saw such attentional scaling of behavioral tuning curves in this same task[37], but more work is needed to demonstrate this role for DR-5HT in our attention task and define which cortical areas experience such scaling effects.

## How could DR-5HT influence attentional networks?

The normalization model of attention captures observations that potential targets compete for attentional enhancement[44]. In this scheme, the brain biases one target's representation (attention field), which helps it compete with other target representations (normalization) and win limited cortical resources[5–9,66]. Since our modified normalization model inherits this logic and regulates normalization with DR-5HT (Fig. 6b–e), it suggests that 5HT may influence such competitive interactions.

While the normalization model does not predict specific attentional circuitry[44], elegant work in non-human primates indicates that target representations are distributed across a network spanning posterior sensory, parietal, and frontal cortical areas[1–10]. Since competition among representations is predicted to occur in every area in this network, the role of DR-5HT control signals should also be similarly distributed. The broad innervation of cortex by DR neurons[27–31], could be an important clue in this regard.

While evidence for a primate-like (fronto-parietal-posterior) attentional networks in mice is still growing, initial optogenetic studies suggest some similarities. For example, stimulation of frontal cortical regions like anterior cingulate and secondary motor cortex can excite V1 neurons[67–69]. Imaging GCaMP-expressing DR boutons or GRAB5HT across the dorsal cortex with new widefield imaging methods[70] in our task could reveal if such areas show attention-related DR-5HT dynamics.

## Neuromodulators and visual attention

William James famously wrote, "Everyone knows what attention is"[71]. Over a century later, we still do not fully understand visual attention in biological terms. While attention's effects on individual neurons and populations have become clearer in recent decades, attention's biological origins remain poorly defined. Stimulation experiments in non-human primates and mice suggest that frontal cortical feedback to posterior visual areas is an important contributor to attentional enhancements. However, a separate body of work suggests that subcortical neuromodulatory areas are a major uncharacterized source of attentional modulation[11–14]. Our observations support and extend this view.

We saw that rapid DR-5HT dynamics regulated attentional strength. Our measure of attentional focus was unaffected by these dynamics, which indicates that this part of the attention computation (i.e., focus) resides elsewhere. Frontal cortical areas are likely important for attentional focus, given reports that their stimulation excites visual cortical areas in ways that mimic the effects of attention in non-human primates[72–78] and mice[67–69]. Interestingly, such frontal cortical areas also represent the strongest inputs to the DR[27–31]. Retrograde tracing from DR and visual cortical areas could reveal whether V1 and DR receive input from shared or unique frontal populations.

The inverse relationship between DR-5HT and attentional strength may also mean another signal, with a non-inverted relationship to attentional strength, balances DR-5HT's effects. Such potential push-pull dynamics offer rapid control, and candidate factors could be norepinephrine[13,51–55] and/or acetylcholine[79–81]. Studying these candidates with our task and behavioral methods could be a way to address this issue.

Finally, a small set of studies suggests that 5HT influences human attention[82–87]. Studies that depleted 5HT and showed improved attention[82] and those showing weakened attention after use of hallucinogens that act as 5HT receptor agonists[87] are consistent with the data we have shown here. Screening the many drugs that target 5HT signaling in our task could reveal new routes to improve the altered attention that characterizes several disorders, including attention-deficit hyperactivity disorder and autism[88].

## Methods
### Animals
Male and female Cdh13-CreER (Cdh13[Ce]), ePet1-Cre (ePet1), Ai95, and Ai27 mice were used in this study. Mice were housed in the Goodman Cancer Research Center Vivarium on a 12 h light/dark cycle using standard room temperature and humidity requirements. Mice for behavioral studies were reverse cycled. Rosa26-LSL-ChR2-tdTomato (AID27, RRID:IMSR_JAX:012567) and Rosa26-CAG-GCaMP6f (RRID: IMSR_JAX:028865) mice were purchased from the Jackson Laboratory and crossed with ePet1-Cre mice or Cdh13-CreER for recording and optogenetic experiments, respectively.

We injected tamoxifen to drive selective expression in Cdh13-CreER (i.e., GCaMP, ChR2, Jaws). Cre expression in the ePet1 line was constitutive. All surgical and experimental procedures were in accordance with the rules and regulations established by the Canadian Council on Animal Care, and protocols were approved by the Animal Care Committee at McGill University.

### Tamoxifen
Tamoxifen (TMX) was dissolved in anhydrous ethanol at 200 mg/mL, diluted in sunflower oil to 10 mg/mL, sonicated at 40 °C until dissolved, and stored at −20 °C as previously described[89–91]. Prior to injection, TMX aliquots were heated to 37 °C and a single injection was delivered intraperitoneally at ~1 g/50 g body weight to Cdh13-CreER -P30 mice.

## Behavioral arena

We used the same behavioral arena as described previously[37]. Briefly, the behavioral apparatus consisted of a custom-built soundproofed dark box that displayed visual stimuli via two 60 Hz LCD screens displayed at a 32° angle from the mouse's midline. The mouse was head fixed in the center of the apparatus on a platform, and a lick spout was positioned to administer liquid reward. Licking was capacitively sensed and digitized using custom electronics. Liquid reward delivery was controlled with a solenoid pinch valve driven by custom electronics. The behavioral task and presentation of visual stimuli were controlled using MATLAB's psychophysics toolbox (Psychtoolbox3) and custom data acquisition software.

## Behavioral task design

We trained mice in our task as described previously[37]. Briefly, mice performed a spatially cued detection task and were trained to lick in response to a 3-bar grating embedded within a noisy checkerboard background. The grating coherence was parametrically controlled by altering the number of noisy checkers that the 3-bar grating contained (Fig.1b). At a grating coherence of 1, the 3 bars contain no noisy checkers, although checkers surrounding the 3 bars are always noisy. At a grating coherence of 0, all checkers are pure noise. While the 3 vertical white bars in the grating increased vertical and local contrast energy, the total number of white checkers in the checkerboard stimulus was always 50%.

Trials began with a sided auditory cue and a static random checkerboard. Mice would lick to start the dynamic checkerboard noise. The coherent grating then occurred on the cued screen at a random time, chosen between 3 and 12 s. For each individual trial, the wait time before the coherent grating appeared was randomly chosen to produce a constant probability of appearing[37] (i.e., a flat hazard function).

A lick that occurred within 0.1–1.5 s following the 3-bar grating was labeled as a hit and led to rewards. The absence of licks during this window were misses and was not rewarded. If mice licked before the onset of the 3-bar grating (false alarm), the trial was restarted. Restarts caused the audiovisual cue to reappear for 0.5 s, followed by a return of dynamic checkerboard noise.

## Training schedule

Reverse-cycled (12 h/12 h inverted light-dark) mice were trained as previously described[37]. Briefly, water-restricted mice were habituated to behavioral arenas for 2 days prior to behavioral shaping. Our shaping protocol is as follows: (1) 1 day to associate licking with liquid reward; (2) 4 days to associate strong 3-bar gratings (coherence > 0.8) with reward; and (3) 7–10 days to lengthen maximum wait times to 12 seconds and decrease grating coherences. By session 15, mice produced reliable psychometric curves. A typical training session lasted 1.5–2 h. Mice were never rewarded for false alarm licks.

## Behavioral analysis of grating detection

For photometry experiments, we pooled trials from later sessions (>session 15) and divided them into DR-low and DR-high categories before computing psychometric measures of performance. For optogenetic studies, control datasets consisted of 10–15 sessions prior to LED stimulation; experimental datasets consisted of all trials during LED stimulation.

Psychometric functions for hits as a function of grating coherence were described by a Weibull function, which provided four parameters: mean, threshold, response bias (lower asymptote), and lapse rate (higher asymptote). Here, threshold represents the grating coherence that is detected at 50% rate of the total response range. Signal detection theory was used to compute the discriminability index d-prime = $Z(PH) - Z(PH0)$ and criterion = $-0.5X(Z(PH) + Z(PH0))/$ d-prime, where $PH$ is the probability of a hit at a single grating coherence. $PH0$ is the probability of a lick to a zero-coherence grating, and Z is the inverse of the standard normal cumulative distribution function. For example, if a mouse were to disregard the visual stimulus and randomly lick, then PH0 and PH would be the same and d-prime = 0 for all coherence levels.

Reaction times were described by a standard exponential function: $ae^{-b(Coherence)} + c$, with amplitude ($a$), baseline ($c$), and slope ($b$).

## Quantification of behavioral sensitivity and attention

Our previously published logistic model revealed that attention multiplicatively scaled mouse behavioral sensitivity for grating-like features on the cued versus the uncued side[37]. Our current model simplifies this approach by capturing this attentional scaling in a single parameter. Briefly, we used false-alarm licks preceded by at least 1–3 s of lick-free random checkerboard noise from the final 20 sessions of mice. This removed FA lick "bouts" and FA licks occurring before the reaction time window but after grating presentation.

Features were computed as follows: Vertical stimulus energy was computed at the same spatial frequency as the target stimulus (0.077 cycles per degree) by convolving quadrature phased Gabor filters (26 x 26 checkers, S.D. = 6 checkers or 3° of visual angle) with each stimulus frame. Contrast energy was computed by convolving each stimulus frame with a Gaussian filter (S.D. = 4 checkers, 2°).

Resulting energy maps for the two features were then down sampled in time from 30 Hz to 10 Hz and spatially down sampled from 17 x 26 checkers to 3 x 5 checkers to ease computational load. Energies for both features were $z$-score normalized.

FA licks were modeled using the logistic function:

$$P_L = \frac{1}{1 + e^{-\sum_f \beta_{f, xy} (\alpha_{Cued} E_{Cued, f}(t-\delta) + \alpha_{Uncued} E_{Uncued, f}(t-\delta)) - k}} \quad (1)$$

In this equation, the probability of a FA lick ($P_L$) is a function of z-scored stimulus energy features ($E_{Cued, f}$ and $E_{Uncued, f}$), where ($f$) is the vertical or local contrast. In this model, $\delta$ is the time it takes for stimulus energy to produce a FA lick, chosen as a hyperparameter based on each individual mouse's psychometric reaction time distribution; we previously showed individualized reaction times for mice[37]. We optimized 3 unknown model parameters: $\beta_{f,xy}$ is the sensitivity of FA licks to the stimulus energy on both the cued or uncued screen for each individual checker at location ($x, y$) for each feature ($f$); $\theta$ is the gain parameter (constrained to [0, $\pi/2$]) which determines the weighting of cued and uncued stimulus energy ($E_{Cued, f}$ and $E_{Uncued, f}$), and $k$ is the weighted stimulus energy corresponding to $P_L = 0.5$. The weighting of cued and uncued energies was captured by the relationship:

$$\alpha_{Cued} = \sin(\theta) \text{ and } \alpha_{Uncued} = \cos(\theta) \quad (2)$$

(also see Fig. 1d).

The model parameters can be interpreted as follows: $\beta$ is a spatial distribution of behavioral sensitivity applied to both the cued and uncued screens, and $\theta$ is the gain parameter that scales energies from the cued side if $\theta > \pi/4$, such that $\alpha_{Cued} > \alpha_{Uncued}$. Both $\beta$ and $\theta$ were optimized to maximize the model predictive power over FA behavior (Supplementary Fig. 1a). Gain $\theta = pi/4$ means no bias to cued features over uncued features, and $\theta = pi/2$ is 100% bias to cued features over uncued features.

We treated the model as an optimization problem for a constrained nonlinear multivariable function, implemented using MATLAB's *fmincon* with the negative-log-likelihood cost function:

$$NLL = \text{argmin} \frac{1}{n} \sum_{i=1}^{n} -[\log(\sigma(y=1)) + \log(1 - \sigma(y=0))] - \lambda \sum w^2 \quad (3)$$

where λ is the regularization factor for standard L2 regularization, $w$ are the parameters $\beta_{x,y}$, $y$ are the $n$ observations (lick or no lick), and σ the standard logistic function. Each model was 5-fold cross-validated to optimize hyperparameters without bias. Model predictive power was measured using the area under the receiver-operator-characteristic.

## Normalization model of attention

To understand how DR-5HT may contribute to suppressive drive in the normalization model of attention[44], we modified the model (http://www.cns.nyu.edu/heegerlab/) directly by introducing a single parameter that multiplicatively scales suppression drive, such that:

$$R = \frac{E}{\epsilon S + \sigma} \quad (4)$$

where $R$ is the population response, $S$ is the suppressive drive, $E$ the excitation field, $\sigma$ the constant that determines contrast gain. DR-5HT low was modeled $\epsilon = 0.0002$ and $\sigma = 3 \times 10^{-7}$, with other parameters adopted from the original model code. DR-5HT high was modeled with the exact same parameters except $\epsilon = 0.003$, thereby increasing the effective suppressive drive (Fig. 6).

## Fiber optic implants

For photometry and optogenetic studies, mice were anesthetized (isofluorane, 4% induction, 1–2% maintenance), mounted on a stereotaxic frame and eye ointment was applied to prevent corneal drying. Following lidocaine application and subcutaneous carprofen injections, a circular incision was made, and the underlying fascia cleaned to dry the skull. Bregma and lambda were then levelled to within 0.1 mm of one another. A 0.1 mm craniotomy was performed above the DR, and the fiber implant (400 μm diameter fiber, Thorlabs) was slowly inserted using a stereotactic arm (4.7 mm posterior from bregma, on the midline and 3.1 mm below the pia). Metabond cement was applied around the implant location and a headplate was cemented additionally to the skull. Any remaining exposed skull was sealed using cement. Following surgery, the animal was removed from the stereotaxic frame and placed on a heat pad for recovery, and analgesia was administered subcutaneously following surgery for 3 days. Shams received implants but did not express optogenetic constructs.

## Cranial window

For GRAB5HT imaging in V1, mice were anesthetized and mounted on a stereotaxic frame as described above but received a 5 mm diameter craniotomy over the left V1 followed by a circular glass coverslip (4 mm diameter) centered 1 mm rostral from lambda and 2.5 mm lateral from the midline. Cranial windows were fixed using instant glue and Metabond at a 15⁰ angle to the mediolateral axis and mice then received a metal headplate implant that surrounded the craniotomy.

## Injections

AAVs were intracranially injected using previously described methods[91,92]. Mice used in the optogenetic suppression experiments received a 1 μl injection of AAV1-CAG-FLEX-rc [Jaws-KGC-GFP-ER2] in the DR (4.7 mm posterior from bregma, on the midline and 3.3 mm below the pia). Mice for 5HT imaging received three 500 nl infusions of AAV9-hsyn-GRAB_5-HT1.0 in the left visual cortex (centered around 1 mm rostral from lambda, 2.5 mm lateral from the midline at a 30⁰ injection angle and 0.2 mm below the pia). A microsyringe pump (UMP3-4, World Precision Instrument, Sarasota, FL) was used to infuse the virus at 5 nL/s.

## Viruses

AAV9-hsyn-GRAB_5-HT1.0 (Cat# 140552-AAV9) and AAV1-CAG-FLEX-rc [Jaws-KGC-GFP-ER2] (Cat# 84445-AAV1) were purchased from Addgene and used for intracranial injections.

## Photometry

A 1-site custom-built fiber photometry system was used to assess changes in GCaMP6f fluorescence from DR neurons. Calcium-insensitive fluorescence was measured with 405 nm light oscillating at a carrier frequency of 450 Hz. Calcium-sensitive fluorescence was measured with 472 nm light oscillating at 205 Hz. An optical fiber (Thorlabs) was used to both deliver and collect light from the brain. Light emitted in the 500–550 nm band was measured using a femto-watt photoreceiver and digitized 200 Hz with an RX8 signal processor (Tucker Davis Technologies). The recorded signal was demodulated according to the two carrier frequencies and low-pass filtered at 20 Hz. Calcium signals were discretized into individual sessions, and calcium-insensitive recordings were fit to the calcium-sensitive signal using least-squares regression using a 5-min moving window. The fitted calcium-insensitive signal was subtracted from the total signal (Supplementary Fig. 1c, d). The resulting signal was normalized to the mean of each session via $z$-scoring to yield a final measure of ΔF/F.

## Wide-field imaging through the cranial window

We used a custom-built epifluorescent microscope to image through our cranial windows with a 10 Hz sampling rate at 512 x 512 pixel resolution. Since windows were mounted at a 15° angle with respect to the mouse's medio-lateral head axis, we mounted the microscope at the same angle such that the mouse heads remained leveled with respect to the visual stimuli. GRAB5HT fluorescence was measured with a 472 nm excitation LED and LabVIEW custom code. Session signals were $z$-scored pixel-wise to yield a final measure of ΔF/F. Signals in Fig. 5 are pooled averages of the brightest 10% of pixels in the image.

## Photometry signal processing

To explore how DR activity on short timescales affected animal performance in our detection task, we first high-pass filtered our ΔF/F session signals recorded via fiber photometry or widefield imaging with a cutoff frequency of 1/40 Hz (Supplementary Fig. 1c–e). Since only hit trials contained reward events that accompanied large positive DR reward signals (see Fig. 2f, g), a filter may artificially introduce baseline shifts preceding the grating presentation on hit trials, but not on miss trials. To eliminate these potential baseline artifacts preceding grating presentation, we removed reward signals from our data before filtering (Supplementary Fig. 1g). Doing so ensured that hits and misses were treated the same. We then computed the average baseline activity in a window preceding grating onset for each trial; this time window was half the length of the unique delay period for each trial. Trials below the median baseline activity of all trials were categorized as DR low, trials above the median were categorized as DR high. We further excluded 5% of the trials closest to this median to ensure the absence of baseline-changing artifacts. False alarms were similarly grouped into DR low and DR high based on the mean activity of 1.5 s preceding the false alarm.

For long-time-scale analysis (Supplementary Fig. 3), session signals were low-pass filtered using a 5 Hz cutoff frequency, and a subsequent rolling mean with a window length of 1 min was applied. Trials in the session were then identified as DR low if most of the trial signal (80% of data samples) remained below the median of the session signal. Trials above the session signal were labeled as DR high.

## Optogenetics

We developed custom code on an Arduino Due to drive a high-powered LED (Thorlabs) that was attached to the mouse's implant through an optical fiber (400 μm diameter, Thorlabs). Our custom behavioral code was adjusted to have full control of the LED stimulation time. For experiments involving mice expressing ChR2, we drove a 470 nm LED at 20 Hz (150 ms pulse duration, 17.2 mW). Mice carrying AAV1-Jaws received a constant 595 nm LED input (8.7 mW) for the duration as suggested previously[93]. Mice were trained fully on the

visual detection task for 30–40 sessions, the last 10–15 of which constituted the control sessions (see Ctrl, Fig. 4e–n). Trained mice then received LED stimulation on every trial for at least 15 more sessions. LED stimulation began 2 s before the grating stimulus and was scheduled for a duration of 3 s. False alarm licks interrupted ongoing LED stimulation and led to trial restarts as described above.

## Histology

Mice were euthanized by isoflurane overdose and perfused intracardially with phosphate buffered saline (PBS), followed by 4% paraformaldehyde (PFA) dissolved in PBS chilled to 4°. Brains were removed and postfixed in 4% PFA overnight. Next, brains were embedded in 2.5% agarose (Sigma), mounted on a tissue slicer (Compresstome), and 150 um thick coronal brain sections were sliced and stored in PBS until immunostaining. Brain sections were incubated with blocking buffer (10% normal donkey serum, 0.4% Triton X-100 in PBS for 1–2 h), then incubated for 7 days at 4 °C with primary antibodies, and with secondary antibodies overnight at 4 °C. Primary antibodies were used as follows: rabbit anti-TPH2 (1:1000, Novus; 1:500, Synaptic Systems); chicken anti-GFP (1:1000, Abcam). Secondary antibodies were conjugated to Alexa Fluor 488 (Cederlane) or Cy3 (Millipore). Sections were mounted onto glass slides using Vectashield (Vector Lab) or Fluoromount (Millipore) mounting medium.

All antibodies were used at dilutions recommended by the manufacturer: Rabbit anti-TPH2 (Polyclonal, Novus, 1:1000, Cat. No.: NB100-74555); Rabbit anti-TPH2 (Monoclonal, Synaptic Systems, 1:500, Cat. No.: 348 003); Chicken anti-GFP (Polyclonal Abcam, 1:1000, Cat. No.: ab13970); Donkey anti Chicken Alexa Fluor 488 (Cedarlane, 1:1000, Cat. No.: 703-545-155); Donkey anti Rabbit Cy3 (Polyclonal, Millipore, 1:1000, Cat. No.: AP182C); Experiments were performed using a single lot per antibody.

Immunostained images were acquired from a Zeiss confocal microscope, using 405, 488, 559, and 635 nm lasers. ImageJ (NIH) software was used to analyze confocal stacks and generate maximum intensity projections to one image.

## Reporting summary

Further information on research design is available in the Nature Portfolio Reporting Summary linked to this article.

## Data availability

Datasets from the current study are available from the corresponding author upon request. Data used for figures is available in the source data file. Source data are provided with this paper.

## Code availability

Some custom code was used for behavioral analysis. It is available at https://github.com/Swamylab/Lehnert_et_al_5HTAttention.git. Example data from one mouse is found at https://doi.org/10.5061/dryad.5qfttdzms. Additional data from other mice are available from the corresponding authors upon request.

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

## Acknowledgements

We thank A. Rangel Olguin, N. Brake, and F. Jalondoni for helpful discussions and contributions. This work was supported by grants from the Scottish Rite Charitable Foundation (A.Kr.), the New Frontiers in Research Fund (A.Kr. and E.C.), Natural Science and Engineering Research Council of Canada (A.Kr. and E.C.), Canadian Institutes of Health Research (A.Kr. and E.C.), and a Canada Graduate Scholarship to J. Lehnert.

## Author contributions

J.L., E.C., and A.Kr. conceived of this study and aided with instrumentation, analysis, experiments, writing, and making figures. J.B. provided the photometry system and gave experimental advice. JL and KC devised the behavioral model with guidance from E.C., A.Kh., and A.Kr. K.Y., J.F., S.S., and X.M. helped with behavioral studies.

## Competing interests

The authors declare no competing interests.
