## [Transparent Peer Review File · Nature Communications]

Rapid dynamics of dorsal raphe serotonin neurons regulate the strength of visual attention

Corresponding Author: Dr Arjun Krishnaswamy

Version 0:

Reviewer comments:

Reviewer #1

(Remarks to the Author)

In the article "Rapid dynamics of dorsal raphe serotonin neurons selectively regulate the gain of visual attention", Lehnert and colleagues are investigating the role of the Dorsal Raphe serotonergic (DR-5HT) neurons in the mouse performance at a visual detection task. In the task, mice are placed head-fixed in front of two screens covering the right and left visual field respectively. An audiovisual cue indicates on which screen the rewarded cue will be presented. Then, a dynamic checkerboard noise is shown. The rewarded stimulus (a 3-bar vertical grating) is presented at a random time during this noise, and the mouse must lick to obtain the water reward. The authors find that the mouse will be more likely successful at detecting the rewarded stimulus presented during the checkerboard noise when DR-5HT neurons are less active. Using a model, they found that false alarms (FA) are more likely to be triggered by events of the checkerboard noise that could look like the rewarded stimulus (enhanced vertical and contrast energy). Defining attention is tricky and varies from lab to lab. Here, attention is defined as a modulation of the gain of neurons of the primary visual cortex (V1) encoding for the rewarded cue. The main hypothesis of the authors is that the DR-5HT neurons modulate the gain of V1 neurons in the attended visual field. This idea is very interesting, yet some important elements of the demonstration are missing to be convincing.

1. Laterality. The claim that DR-5HT activity supports attention in V1 is supported by a model that shows that false alarms (FA) are more likely to occur when events in the checkerboard noise that are compatible with the rewarded cue by their vertical and contrast energy are presented on the screen that was cued as being the one where the rewarded stimulus will be presented. It is to be noted that the predictive power of the model is low (the area under ROC is between 51 and 56%, Ext data Fig., 1). Yet, following the hypothesis that DR neurons modulate the gain of the V1 neurons that are expected to provide the behaviorally relevant information (i.e., neurons located in the V1 contralateral to the cued screen), we could expect to see that the neurons of the right and left hemisphere DR would be differentially active when the cue is indicating that the grating will be shown on the right or the left screen respectively. Yet the fiber photometry recording in the study averages the activity of all the DR neurons. The authors are simultaneously recording the neurons of the right and left DR. Therefore, it is not clear if the DR activate the V1 corresponding to where the rewarded stimulus will be presented or just provide the same nonspecific signal to the two V1. Therefore, I would be interested in seeing the difference in serotonin release in V1 (experiment related to Fig. 5) when the task cue indicates that the stimulus will be presented in the ipsi- and contralateral side, and to perform recording with a single cell resolution in DR to determine the diversity of activity in DR neurons. Showing that DR neurons targeting V1 and frontal cortices present different type of activity during the task would be the best.

2. Specificity. The study is mainly interpreted through the prism of a modulation of the gain of V1 neurons. Yet, the activity of the neurons in V1 is not recorded. The evidence of a modulation of the gain of V1 neurons and how this modulation favors the response of V1 neurons to noisy stimuli that are compatible by their vertical and contrast energy with the rewarded cue during FA needs to be provided. Recording the neurons in V1 during the task and determine if the gain mechanism hypothesized by the authors can be confirmed experimentally would considerably strengthen the study. Moreover, to demonstrate the specificity of the DR-5HT activity in V1 (i.e., that the putative change in the gain of V1 neurons is not indirectly due to a DR-5HT modulation of the activity of neurons of the frontal areas providing feedback to V1), the authors should measure the release of 5HT in the frontal cortex. The anterior cingulate cortex could be a good target as it was showed to play a role in spatial attention in V1. If some DR-5HT modulation was to be found in the prefrontal cortices, the

authors would have to demonstrate that the changes in detection sensitivity cannot be explained by the top-down information provided to V1 by those frontal areas.

3. Slow versus fast DR-5HT dynamics. The authors show that there are two kinds of dynamics in the DR-5HT activity. A slow component (Ext Data Fig. 3) and a fast component (Fig.2). Are those two components also found in the release of 5-HT in V1? As those two components seems to have the same effect on detection, I would assume that a short decrease of DR-5HT as in Fig.2g is less effective when it occurs in the rising phase of the slow fluctuation (like for the Hit trial in the inset of Ext Fig. 3a). Is it the case?

4. Locomotion and arousal. Locomotion (and movement in general) as well as arousal (typically measured by the pupil size) have been shown to play a major role in modulating the gain of V1 neurons. It would be therefore important to demonstrate that the results of the authors cannot be explain by those factors.

Minor points.

How was the D prime computed? What maximal D prime can be obtained by a mouse performing the task at random? Gratings switch sides every 35 trials. Isn't this rule undermining the role of the cue indicating where the rewarded stimulus will be presented? Can mice perform the task if the rewarded stimulus is randomly presented on the right and left screen?

Reviewer #2

(Remarks to the Author)

Lehnert et al present a very nice study revealing in mice that fast activity of dorsal raphe serotonergic neurons selectively regulate attentional gain but not spatial allocation in a visual attention task. The authors used a cleverly designed cued visual detection task that was previously developed by the lab to detect 3-bar gratings embedded in zero-coherence dynamic checkerboard noise. Such task design allows authors to use vertical and contrast energy in the visual stimuli and behavioral performance to compute attentional gain the allocation. With the task, the authors used optic approaches to record and causally manipulate (both activate and inhibit) neuronal activity in dorsal raphe nuclei, and found that increased peri-detection raphe activity correlated and causally contributed to reduced attentional gain, whereas decreased raphe activity contributed to elevated attentional gain. Crucially, they found that contribution of 5-HT on attention was specific to attentional gain control, rather than alter its spatial allocation or other aspects of attentional performance, e.g. lapse rate, criterion or reaction time. Lastly, they found in visual cortex V1 5-HT low or high release preceding grating appearance correlated high or low task performance respectively, resembling their DR photometry data. The study provides crucial mechanistic insights on how fast dynamics of 5-HT release regulate visual attention. The experiments were soundly designed and conducted, data analysis was rigorous, and text was well-written. Overall, I really like the study and enjoying reading the manuscript. I believe the study will generate great interests in the entire vision community. I only have a few minor comments that I would like authors to clarify to improve the manuscript.

My specific comments:

1. The conclusion that 5-HT modulates attentional gain but not focus was based on attention maps constructed with vertical and contrast energy in the visual stimulus and FA licks during different 5-HT levels or manipulation conditions. I feel this might be a bit overstated, given the task design and data shown. In the task, the cue was 100% valid and the 3-bar gratings were always on the center of the cued screen. Therefore, it seems to be a given that high sensitivity to vertical or contrast energy remained to be at the center of the screen regardless of 5-HT levels. What could be other potential scenarios in the attention map? Perhaps authors could instead look into a subset of trials at the beginning of cue/block switch, where spatial predictiveness of the target is at the lowest to reveal whether 5-HT level is indeed involved in attentional allocation.

2. The visual stimuli were lateralized in the task. Was the involvement of 5-HT in attentional modulation lateralized? Photometry and optogenetic stimulation in DR in the study appear to cover both hemispheres, however, 5-HT wide-field imaging in V1 was done unilaterally. Does the predictiveness of V1 5-HT in attention performance depend on laterality of the cue?

3. The authors appear to fixate that 5-HT modulation on visual attention is through V1. Indeed, authors found 5-HT dynamics in V1 resembled that of DR photometry recordings. However, such correlation was not causal. Optogenetic stimulation of DR can affect 5HT release in many other downstream targets. Besides V1, DR-5HT neurons also densely project to superior colliculus (SC) and exert potent suppression of SC visually evoked activity. Given SC is involved in visual attention, including in mice, it is plausible that 5-HT release in SC is playing an even more prominent role in attention. Ideally, optogenetic manipulation of 5-HT axon terminals in V1 could provide a definitive answer. The authors should at least discuss 5-HT release in other visual areas in attentional control.

Reviewer #3

(Remarks to the Author)

Reviewer #4

(Remarks to the Author)

In this manuscript, Lehnert et al. show that 5-HT modulates visual attention, with decreases in 5-HT neural activity and

release in V1 resulting in greater attention towards visual cues. The authors employ a useful behavioral paradigm for tracking attention in addition to fiber photometry, optogenetics and widefield imaging to track and manipulate DR neurons/5-HT release during execution of this task. Consistent findings across these techniques are a strength of the manuscript. The data could be more clearly presented and described.

Showing fluorescence data without filters applied would be a useful supp figure given the authors noted caveat the filters can change activity around large deviations (even if the authors removed the Hit activity prior to the filter, it would be good to show less processed data that show this to be the case)

Why is the baseline attentional index data for the optogenetic cohorts much lower than either recording cohort? And the attentional index equation/explanation should be included in text not just figure.

Are there differences in behavior and/or 5-HT activity in trials that follow a FA vs trial start lick?

Figure 3 d prime and threshold plots should start at 0 on the y axis.

Minor comments:

-5HT recording data at the level of V1 does not correlate as robustly with attention as photometry data, and gain/loss of function data is at the level of the DR. This limits the ability to state that the 5-HT modulation is at the level of V1 (or at least only at the level of V1).

-In some supp figure legends $\Delta f/F$ is written as $\Delta f/F_f$

-Colors in supplemental figures often re-use blue/orange color scheme despite not referring to the same features as the main figures (e.g., Figure S3 uses blue/orange for hit/miss and has entirely new colors for high/low DR activity). Figures should have consistent themes to help readability.

Version 1:

Reviewer comments:

Reviewer #1

(Remarks to the Author)

I am grateful to the authors for their detailed and thoughtful replies to my initial questions. Despite these clarifications, I still find it difficult to fully grasp the impact and broader meaning of the results presented in this paper. My second round of questions is therefore aimed at achieving a more precise understanding.

Based on the authors' responses, it is now clear that:

1. Scope of the main findings

The primary result of this paper is not a modulation of visual processing by serotonin within visual cortex (as I originally suspected in Question 1), but rather a more general effect of serotonin on visual detection behavior. The authors' data do not support any claim of a gain change in V1, even if such a mechanism remains possible. Indeed, as they note in the first paragraph of the discussion section ("How does DR-5HT contribute to attentional network"), serotonin's action on V1 networks could contribute to attentional gain modulation. Given that serotonin is generally believed to suppress cortical excitatory neurons, a decrease in serotonin release within V1 would be expected to increase neuronal excitability.

2. On lateralization

My second concern stemmed from the apparent lateralization of attention to the visual stimulus, which initially raised the possibility that serotonin might contribute to lateralized visual processing. However, the authors' preliminary results do not support this view: no evidence of lateralization in serotonin-related activity was observed. The source of this lateralization therefore remains unknown and is unlikely to be directly explained by lateralized serotonergic modulation. As the authors themselves note in their reply to Question 2, serotonin is more likely involved in setting a general attentive brain state, albeit in a negative fashion (see comment 3 below). In this context, the observation that excitation of DR-5HT neurons is associated with a switch of attention to the unattended side (Fig. 4i) is difficult to reconcile with their interpretation. This finding is neither highlighted in the results section nor addressed in the discussion.

3. Interpretation of DR-5HT suppression

A central aspect of the narrative strikes me as counterintuitive across the manuscript. The authors present the suppression of DR-5HT activity during successful sensory processing as evidence for an active involvement in attention. Yet, DR-5HT neurons are less active during these hit trials. Their activity instead increases during reward processing. The paper frames this suppression as an "active implication," which risks inverting the logic of interpretation. To illustrate: it is as if a newspaper were to write, "Traffic accidents regulate improvements in city traffic," when in reality accidents hinder traffic flow rather than drive its improvement.

Taken together, the key finding is that DR-5HT activity is suppressed immediately before hit trials in a visual detection task. Yet, without a clear neuronal mechanism linking this activity to behavior, the broader relevance of the study for the neuroscience community remains limited.

Reviewer #2

(Remarks to the Author)

In the revised manuscript, with all the changes and the additional analysis provided, all my previous concerns now have been addressed. I have no additional concerns, and I support the publication of the study.

Reviewer #3

(Remarks to the Author)

The authors have adequately addressed all my comments.

Reviewer #4

(Remarks to the Author)

The authors have addressed several concerns.

-Given that several reviewers misunderstood the FA/start lick structure it would be useful to better communicate this in the manuscript rather than just in the response to reviewers.

Version 2:

Reviewer comments:

Reviewer #1

(Remarks to the Author)

I am very grateful for the time and energy the authors dedicated to addressing my questions. The paper is much clearer in its current version regarding the underlying hypotheses, findings, and conclusions. I have no further questions.

Response to reviewer 1

In the article “Rapid dynamics of dorsal raphe serotonin neurons selectively regulate the gain of visual attention”, Lehnert and colleagues are investigating the role of the Dorsal Raphe serotonergic (DR-5HT) neurons in the mouse performance at a visual detection task. In the task, mice are placed head-fixed in front of two screens covering the righty and left visual field respectively. An audiovisual cue indicates on which screen the rewarded cue will be presented. Then, a dynamic checkerboard noise is shown. The rewarded stimulus (a 3-bar vertical grating) is presented at a random time during this noise, and the mouse must lick to obtain the water reward. The authors find that the mouse will be more likely successful at detecting the rewarded stimulus presented during the checkerboard noise when DR-5HT neurons are less active. Using a model, they found that false alarms (FA) are more likely to be triggered by events of the checkerboard noise that could look like the rewarded stimulus (enhanced vertical and contrast energy). Defining attention is tricky and varies from lab to lab. Here, attention is defined as a modulation of the gain of neurons of the primary visual cortex (V1) encoding for the rewarded cue. The main hypothesis of the authors is that the DR-5HT neurons modulate the gain of V1 neurons in the attended visual field. This idea is very interesting, yet some important elements of the demonstration are missing to be convincing.

We are happy the Reviewer finds the central idea of our paper interesting. Before we respond to the reviewer’s individual points, however, we feel that a clarification of our use of the term “gain” is in order. In our study we modeled the *behavioral* allocation of attention using multiplicative gain terms (α_{cued} and α_{uncued} in **Manuscript Fig. 1d**) and found DR-5HT was linked to changes in these

parameters. Our behavioral gains (α), however, are not intended to describe the neural mechanism of attention in V1 or other brain areas. Thus, our results make no claims on whether DR-5HT directly affects affect neural gain at the level of V1. We recognize that our use of the term "gain" to describe the behavioral effects of attention lacked clarity because changes in the gain of neural tuning is one of the mechanisms by which attention exerts its effects. We have now emphasized this point throughout the manuscript and changed our title.

Rev1.1) Laterality. The claim that DR-5HT activity supports attention in V1 is supported by a model that shows that false alarms (FA) are more likely to occur when events in the checkerboard noise that are compatible with the rewarded cue by their vertical and contrast energy are presented on the screen that was cued as being the one where the rewarded stimulus will be presented. It is to be noted that the predictive power of the model is low (the area under ROC is between 51 and 56%, Ext data Fig., 1).

Yet, following the hypothesis that DR neurons modulates the gain of the V1 neurons that are expected to provide the behaviorally relevant information (i.e., neurons located in the V1 contralateral to the cued screen), we could expect to see that the neurons of the right and left hemisphere DR would be differentially active when the cue is indicating that the grating will be shown on the right or the left screen respectively. Yet the fiber photometry recording in the study averages the activity of all the DR neurons. The authors are simultaneously recording the neurons of the right and left DR. Therefore, it is not clear if the DR activate the V1 corresponding to where the rewarded stimulus will be presented or just provide the same nonspecific signal to the two V1.

The reviewer asks if our recorded DR-5HT signals have a lateral component. In our attempt to simplify the logic of our writing, we may have inadvertently suggested that the DR-5HT attention signal must be laterally implemented in V1. It does not.

We explicitly modelled behavior as a set of spatial sensitivities to vertical and contrast energy (β_v and β_c) and multiplicative gains to cued and uncued screens (α_{cued} and α_{uncued} , see **Manuscript Fig. 1d**). Our results showed that lower activity in the DR and lowered release of 5HT in V1 favored behavioral gain to the cued side. Our results do not address, however, how lowered 5HT release in V1 implements this behavioral change.

The reviewer suggests one possible scenario: a selective reduction in 5HT within the V1 contralateral to the cued screen increases the gain of visual neurons. Another scenario is that lateralized reductions in 5HT could happen in other brain targets relevant to visual attention which receive strong input from DR (ie: frontal cortex, dLGN, etc).

Importantly, the release need not be lateralized at all; DR-5HT could instead gate or increase the efficacy of ongoing neural gain mechanisms that are themselves lateralized. Prior work would favor this latter view because (1) DR-5HT neurons encode internal states rather than external spatial locations¹⁻⁶; and (2) because there is no evidence that DR projections to cortex are lateralized⁷.

We feel there are two major mechanistic clues: (1) recent observations that DR-5HT neurons encode expected uncertainty^{5,6} and (2) older observations suggesting that expected uncertainty tracks cue validity⁸⁻¹³. DR-5HT is low when the outcome (or uncertainty therein) is predicted by the cue and is high when unexpected shifts (eg: block switches in our task, **Reviewer Fig. 8a**) steal predictiveness from the cue. In this scheme, the brain has methods to lateralize visual processing in response to the cue but only does so when the DR-5HT signal deems the cue important.

Whatever the mechanism is, prior work suggests that the spatial and temporal patterns of 5HT release across the cortex are diverse^{3,14-16}. Moreover, the expression of the 14 different 5HT-receptor isoforms across the cortex¹⁷⁻²¹ (including V1) will require significant work to tease apart and relate to the attentional effects we have reported here. We are currently working on this issue,

but to acknowledge the reviewer's interest, we have included some preliminary widefield GRAB5HT mesoscale imaging data from the entire dorsal cortex in **Reviewer Fig. 1**.

Rev1.2) Therefore, I would be interested in seeing the difference in serotonin release in V1 (experiment related to Fig. 5) when the task cue indicates that the stimulus will be presented in the ipsi- and contralateral side,

Briefly, we used established procedures²² to deliver php.Eb AAVs²³ containing genetically encoded 5HT sensors (GRAB5HT^{24,25}) to mice and then mesoscale widefield imaged sensor fluorescence over the entire cortex while mice performed our task (**Reviewer Fig. 1a**). This preliminary widefield dataset confirms our observations from photometry and cranial window imaging (**Manuscript Figs. 2-3,5**) and shows that 5HT release drops in both V1s before hits (**Reviewer Fig. 1b**). Correlating GRAB5HT signal across hemispheres (**Reviewer Fig. 1c, top**) shows that both V1s receive similar but not identical signals (**Reviewer Fig. 1c, bottom**). This

REDACTED

interhemispheric similarity was also observed for many other cortical areas (**Reviewer Fig. 1d-e**). These observations suggest that DR sends a global signal to both visual cortices, thereby enhancing ongoing lateralized mechanisms that embody attentional focus. While circuit-level detail about such focusing mechanisms remains at large, they could involve input from frontal cortical areas described in recent studies^{14,26}.

Rev1.3) ...and to perform recording with a single cell resolution in DR to determine the diversity of activity in DR neurons. Showing that DR neurons targeting V1 and frontal cortices present different type of activity during the task would be the best.

This comment arises from the view that DR-5HT signals are responsible for changes in neural gain of V1. We respectfully point out that we never claimed this. Nonetheless, the reviewer is requesting that we (1) perform cell-type specific extracellular recordings from DR neurons in vivo, and (2) assess whether DR neurons targeting V1 or other areas present different activity patterns.

(1) Type-specific recording of DR diversity: We currently do not have the capability to record from DR neurons; purchase and integration of this equipment into our behavioral setups would take time. This said, there is extensive work on the molecular taxonomy of DR neurons which show 5-7 major classes with at least 10-14 molecularly distinct subtypes^{3,15,27}. These same studies also examined neural activity in select DR-5HT subtypes which together with older work^{1,2,4,16,28,29} suggests a diversity of functional responses across different kinds of DR-5HT neuron. Taken together, this data indicates substantial functional diversity in DR-5HT neural populations.

While we cannot provide extracellular recordings from genetically defined DR-5HT subtypes, our analysis of widefield GRAB5HT imaging during hits already shows some diversity in how 5HT is released across cortex (**Reviewer Figures 1&2**). The rapid drop in 5HT levels that we show regulates behavioral attention occurs mainly in posterior visual and midline frontal areas (e.g. cingulate) before hits. However, we see a rise in 5HT levels over the same period in somatosensory, auditory, and lateral motor regions (**Reviewer Fig. 1d**).

Given this observation, we decomposed our widefield imaging data into its principal components to visualize spatial patterns of GRAB5HT signal that best characterize our data. These components were highly diverse. The top 5 patterns reveal both brain-wide and region-specific patterns of release (**Reviewer Fig. 2**). PC2 is spatially like the pattern seen during hits, whereas PC4 shows that visual areas experience different pattern of 5HT release than the rest of the brain. Taking the results in both **Reviewer Figs. 1-2** together, the spatiotemporal patterns of 5HT release across cortex are diverse and not strongly lateralized.

REDACTED

(2) Do DR cortical projections present different activity patterns? Details about the projection patterns of every DR type is an area of ongoing research. Evidence is limited to a set of recent studies which used injections of retrograde viral tracers in target structures paired with single-cell sequencing in the DR to define how a given DR type projects^{3,15,16,27}. These studies show that cortically projecting types lie directly in the midline under our optical fiber, whereas subcortically projecting ones reside in the lateral parts of DR that are likely poorly sampled by our optical fiber.

An elegant study by Ren et al. 2018, showed that DR types projecting to frontal areas only weakly innervate posterior cortical areas³. Our widefield imaging results support this view and show different neural dynamics in the posterior versus the anterior dorsal cortex during hits (**Reviewer Fig. 1d**) and a functionally defined posterior compartment of 5HT release (*PC4*, **Reviewer Fig. 2b**).

In summary, the reviewer asked that we investigate the diversity of DRN neurons using single cell recordings. Based on previous studies by other labs we feel this would be a major undertaking beyond our current technical abilities and outside the scope of our current study. However, our analysis of hits/misses together with our PCA shown above suggests a diversity of at least 4 different cortical compartments of DR-5HT activity: midline frontal, motor, somatosensory, and visual areas (**Reviewer Figs.1-2**). Our lab is in a good position to continue these mesoscale imaging studies in the future to untangle how these different cortical compartments are linked to the DRN (the main source of 5HT in the forebrain) and attention.

Rev1.4) Specificity. The study is mainly interpreted through the prism of a modulation of the gain of V1 neurons. Yet, the activity of the neurons in V1 is not recorded. The evidence of a modulation of the gain of V1 neurons and how this modulation favors the response of V1 neurons to noisy stimuli that are compatible by their vertical and contrast energy with the rewarded cue during FA needs to be provided. Recording the neurons in V1 during the task and determine if the gain mechanism hypothesized by the authors can be confirmed experimentally would considerably strengthen the study.

The reviewer suggests our hypothesis is that DR-5HT signals regulate attention by modifying the gain of V1 neural tuning. As we mentioned above, we recognize a lack of clarity in the presentation of our results and the use of the term “gain”. Our study selectively links changes in DR-5HT to changes in behaviorally defined attentional strength.

Our results show that a decreased DR activity and 5HT release in V1 are related to an increased ‘behavioral gain of attention’; captured by cued and uncued model weights (α_{cued} and α_{uncued}). These results do not show how reduced DR-5HT implements this behavioral change. Our measures of 5HT release in V1 (**Manuscript Fig. 5, Reviewer Figures 1-2**) and many prior studies³⁰⁻⁴⁰, suggest that V1 could be an important site of DR-5HT action. We have expanded our discussion of potential sites in the revised manuscript.

Providing conclusive evidence and characterization of the V1 neural correlate of attention in our task will take time. We are actively pursuing this line of research, although it is still in its early stages. To address the reviewer’s interest, we show some of our preliminary results in **Reviewer Fig. 3**.

Briefly, we expressed genetically encoded calcium indicators (GCAMP6f) across the cortex and widefield imaged neural signals from the cortex while mice performed our task (**Reviewer Fig. 3a**). Next, we extracted visual contrast energy from noise periods between false alarm licks and correlated contrast energy fluctuations at each checker position to the corresponding movies of cortex-wide GCaMP6f activity (**Reviewer Fig. 3b**). As expected, contrast energy correlations were strongest in the visual cortex (dark spot in **Reviewer Fig. 3b**). The correlations between contrast energy and V1 calcium signals also exhibited the expected retinotopic organization (**Reviewer Fig. 3c**). Interestingly, our early analysis of these correlations when segmenting our data into cued and uncued checkers and brain maps reveals stronger correlations to cued contralateral contrast

REDACTED

energy fluctuations (**Reviewer Fig. 3d**). This example shows that left V1 had a much stronger correlation (dark) with contrast energy when the cue was on the right, while the effect of cue location on the right V1 was more mixed.

These preliminary results indicate that visual cortex can be more strongly driven by cued checkerboard noise than by uncued noise. One interpretation is that the response gain of V1 is enhanced by the cue. As the reviewer is certainly aware of, however, an increased correlation between the neural response and noise stimulus shown in **Reviewer Fig. 3d** does not conclusively demonstrate a multiplicative gain change in V1 (which requires measuring the tuning of V1 responses). Thus, many more experiments are needed to strengthen this result and flesh out mechanisms. Moreover, we plan to explore how this neural correlate of attention varies if we examine other visual features (e.g., orientation energy and spatial frequency), examine its cellular basis with 2 photon imaging, and link these effects to behavioral performance. Should all these stages work out, we would be in a strong position to then study the mechanisms by which 5HT interacts (or influences) these attentional changes to V1.

Although we are actively addressing these interesting and important questions raised by the reviewer, incorporating all these experiments would go far beyond the current scope of this study which is the first to show that DR-5HT signals play a causal role in visual selective attention.

Rev1.5) Moreover, to demonstrate the specificity of the DR-5HT activity in V1 (i.e., that the putative change in the gain of V1 neurons is not indirectly due to a DR-5HT modulation of the activity of neurons of the frontal areas providing feedback to V1), the authors should measure the release of 5HT in the frontal cortex. The anterior cingulate cortex could be a good target as it was showed to play a role in spatial attention in V1. If some DR-5HT modulation was to be found in the prefrontal cortices, the authors would have to demonstrate that the changes in detection sensitivity cannot be explained by the top-down information provided to V1 by those frontal areas.

We respectfully restate that our results make no claim about a gain change in V1. The reviewer requests that we measure DR-5HT release in the anterior cingulate cortex (ACC) to see if this region has the same modulation as we observed in V1. Although we feel this is a question worthy of a separate study, we have generated preliminary results that show the ACC also experiences a rapid, transient reduction in 5HT before grating appearance on hits (**Reviewer Fig. 1d**). To investigate this further, we split our preliminary widefield imaging data into 5HT-high and 5HT-low

REDACTED

groups (**Reviewer Fig. 4a**) as we did for cranial window imaging in V1 (**Manuscript Figure 5**) and then examined the time course of activity across cortical regions before hits and false alarms.

Widefield GRAB5HT imaging confirms our observation of a novel, rapid 5HT release pattern within V1 (**Reviewer Fig. 4b-c**); as in **Reviewer Fig. 1**, and 5HT signals do not appear lateralized to a particular V1. However, segmenting our widefield datasets according to 5HT-low or high release in V1 before grating onset (**Reviewer Fig. 4b**) and false alarm licks (**Reviewer Fig. 4c**) show similar patterns of release in V1, midline frontal areas including ACC, and other areas. This similar 5HT dynamic in ACC and V1 could reflect the strong interconnection between these brain areas and the effects of ACC feedback to V1^{14,26}, including effects that modulate visually guided behaviors²⁶. On the other hand, while similar, ACC and V1 patterns are not identical. For example, 5HT release remains somewhat lower in ACC prior to false alarms even though 5HT release is high in V1 (**Reviewer Fig. 4c**, 5HT-high). More work will be needed to quantify these differences and relate them to attentional and behavioral performance, and we plan to pursue these important questions in the future.

Rev1.6) Slow versus fast DR-5HT dynamics. The authors show that there are two kinds of dynamics in the DR-5HT activity. A slow component (Ext Data Fig. 3) and a fast component (Fig.2). Are those two components also found in the release of 5-HT in V1? As those two components seems to have the same effect on detection, I would assume that a short decrease of DR-5HT as in Fig.2g is less effective when it occurs in the rising phase of the slow fluctuation (like for the Hit trial in the inset of Ext Fig. 3a). Is it the case?

There are three points raised here: (1) The reviewer asks if 5HT release in V1 also has slow and fast component; (2) The reviewer suggests that fast and slow DR-5HT dynamics have the same effects on attention. (3) The reviewer asks if fast DR-5HT low signals are better on the rising phase of a slow DR-5HT signal.

(1) Do cortical GRAB5HT signals show slow and fast dynamics? The slow and fast components that we saw in our DR photometry recordings are indeed mirrored within our GRAB5HT dataset (**Reviewer Fig. 5**). Briefly, raw session fluorescence is detrended (**Reviewer Fig. 5a**) and then either high- or low-pass filtered (**Reviewer Fig. 5b**) to obtain fast and slow GRAB5HT signals respectively (**Reviewer Fig. 5c**).

(2) Do slow and fast DR-5HT dynamics contribute to visual attention? Please note that fast DR-5HT dynamics predict increased behavioral attention gain and performance (**Manuscript Figures 2-3,5**). We now show that the same behavioral measures of performance are unrelated to slow DR-5HT dynamics (**Manuscript Extended Data Figure 3**). We have better clarified this point in our manuscript.

(3) Since behavioral performance was only related to fast DR-5HT dynamics and not slow ones, we predict that this relationship to fast signals will

Reviewer Figure 5. a, Raw GRAB5HT signals from an example behavioral session (top) and detrended signal (bottom). b, High- and low-pass filtering applied to the example detrended session signal in a.

persist regardless of underlying slower dynamics. We nonetheless appreciate the reviewer's curiosity and now provide this analysis in **Reviewer's Fig. 6**.

Briefly, we filtered session signals as we did in our submitted manuscript to identify periods of slow rising and falling baseline periods (occurring over 30 sec-1 min, **Reviewer Fig. 6**). In parallel, we high-pass filtered this signal to identify rapid DR-5HT high and low dynamics (**Reviewer Fig. 6a**). Together, this allows us to analyze behavioral performance of mice during rapid DR-5HT high and low signals in regimes where the baseline was slowly rising or falling. As we showed in **Manuscript Figures 2-3,5** and **Extended Data Figure 3**, performance is only predicted by rapid DR-5HT high and low dynamics. Slow changes in baseline have little or no effect (**Reviewer Fig. 6b-c**). We now include this new analysis in **Extended Data Figure 3** of the revised manuscript.

Reviewer Figure 6. **a**, raw photometry signals from an entire session low pass filtered to identify rising and falling baseline periods (middle) and parallel high-pass filtering to identify fast DR-high and low signals (bottom). **b**, mean psychometric curves of hits and corresponding measures of psychometric threshold, guess rate and lapse rate for DR-high and low trials occurring on a rising baseline. **c**, mean psychometric curves of hits and corresponding measures of psychometric threshold, guess rate and lapse rate for DR-high and low trials occurring on a rising baseline. Slow changes in baseline are not related to detection and behavioral performance ($n=7$ mice).

Rev1.7) Locomotion and arousal. Locomotion (and movement in general) as well as arousal (typically measured by the pupil size) have been shown to play a major role in modulating the gain of V1 neurons. It would be therefore important to demonstrate that the results of the authors cannot be explain by those factors.

There are two points here. First, the reviewer asks about the relationship between locomotion and our attentional and behavioral effects. Second, the reviewer asks whether the effects we saw on attention and behavioral performance could be explained by changes in arousal.

(1) As in our methods, mice sat stationary on a platform during our task and therefore did not run during our experiments.

(2) We did not have the capability to record eye movements in tandem with many of the experiments shown in our original submission. We now have an eye-tracker and have some preliminary eye data in concert with GRAB5HT data. This data is shown in **Reviewer Fig.7**.

Briefly, we recorded a single eye along with widefield imaging GRAB5HT signals from behaving mice. Videos were then processed using facemap⁴¹ to obtain pupil ellipses (**Reviewer Fig. 7a-b**), and then pupil area extracted on hit, ignore, and miss trials (**Reviewer Fig. 7a**). These data show that pupil areas before grating onset on hits and misses were similar, but the pupil areas were slightly smaller on ignore trials, consistent with prior reports suggesting that pupil dynamics correlate with arousal⁷.

We also examined the dynamics between GRAB5HT release in V1 with pupil area across a session (**Reviewer Fig. 7b**). Plotting these traces against one another shows a largely symmetric point cloud with no significant correlation (**Reviewer Fig. 7c**). Taken together, these data suggest that the V1 GRAB5HT dynamics that we linked to attention are likely unrelated to pupil dynamics. We can include some of this data as a part of an Extended Data Figure if the reviewer and editor feel it is appropriate.

Reviewer Figure 7. a, Pupil areas for one mouse taken from the indicated trial outcomes. For hits and misses, pupil areas come from before grating onset. All pupil dynamics were used for ignores. **b**, Example image of a mouse eye with pupil outlined with an ellipse and time-varying changes in pupil dynamics across several trials shown (top). Example widefield image of GRAB5HT fluorescence taken from the same period of time as the pupil dynamics shown at top. Inset shows V1 GRAB5HT dynamics corresponding to the pupil traces. **c**, Scatter plot of pupil area versus V1 GRAB5HT dynamics, take from traces like shown in b.

Minor points.

Rev1.8) How was the D prime computed? What maximal D prime can be obtained by a mouse performing the task at random?

D-prime was computed using both hit rates and false alarm rates. Briefly, signal detection theory was used to compute the discriminability index $d\text{-prime} = Z(PH) - Z(PH0)$ and criterion $= -0.5 \times (Z(PH) + Z(PH0)) / d\text{-prime}$, where PH is the probability of a hit at a single grating coherence. $PH0$ is the probability of a lick to a zero-coherence grating (which provides the false alarm rate for grating detections), and Z is the inverse of the standard normal cumulative distribution function. In our submitted manuscript we compared d-prime at a grating coherence of 0.2.

For a mouse licking at random, we would expect its hit rate to be flat across all coherences. Thus, the probability of a hit at zero coherence ($PH0$) is the same as the probability at any other coherence (PH). Using the equation above returns a d-prime of zero. To improve clarity, we now mention this example in the Methods.

Rev1.9) Gratings switch sides every 35 trials. Isn't this rule undermining the role of the cue indicating where the rewarded stimulus will be presented? Can mice perform the task if the rewarded stimulus is randomly presented on the right and left screen?

Cueing in blocks of trials is standard practice in many studies examining attention in humans⁴²⁻⁴⁷, non-human primates⁴⁸⁻⁵³, and rodents⁵⁴⁻⁵⁸. Moreover, we previously showed that mice use the cue to perform better in this task⁵⁹. Rather than undermining the cue, the block-mode emphasizes

cue importance. We have not explicitly tried randomly presenting stimuli on left and right screens. We assume that in this case, the animals would equally allocate attention to either screen because the average probability of a grating to occur would be 0.5 on each screen on every trial.

Response to reviewer 2

Lehnert et al present a very nice study revealing in mice that fast activity of dorsal raphe serotonergic neurons selectively regulate attentional gain but not spatial allocation in a visual attention task. The authors used a cleverly designed cued visual detection task that was previously developed by the lab to detect 3-bar gratings embedded in zero-coherence dynamic checkerboard noise. Such task design allows authors to use vertical and contrast energy in the visual stimuli and behavioral performance to compute attentional gain the allocation. With the task, the authors used optic approaches to record and causally manipulate (both activate and inhibit) neuronal activity in dorsal raphe nuclei, and found that increased peri-detection raphe activity correlated and causally contributed to reduced attentional gain, whereas decreased raphe activity contributed to elevated attentional gain. Crucially, they found that contribution of 5-HT on attention was specific to attentional gain control, rather than alter its spatial allocation or other aspects of attentional performance, e.g. lapse rate, criterion or reaction time. Lastly, they found in visual cortex V1 5-HT low or high release preceding grating appearance correlated high or low task performance respectively, resembling their DR photometry data. The study provides crucial mechanistic insights on how fast dynamics of 5-HT release regulate visual attention. The experiments were soundly designed and conducted, data analysis was rigorous, and text was well-written. Overall, I really like the study and enjoying reading the manuscript. I believe the study will generate great interests in the entire vision community. I only have a few minor comments that I would like authors to clarify to improve the manuscript.

We thank the reviewer for these very kind words and are delighted that they enjoyed our paper and believe it will generate great interest in the entire community.

My specific comments:

Rev2.1) The conclusion that 5-HT modulates attentional gain but not focus was based on attention maps constructed with vertical and contrast energy in the visual stimulus and FA licks during different 5-HT levels or manipulation conditions. I feel this might be a bit overstated, given the task design and data shown. In the task, the cue was 100% valid and the 3-bar gratings were always on the center of the cued screen. Therefore, it seems to be a given that high sensitivity to vertical or contrast energy remained to be at the center of the screen regardless of 5-HT levels. What could be other potential scenarios in the attention map? Perhaps authors could instead look into a subset of trials at the beginning of cue/block switch, where spatial predictiveness of the target is at the lowest to reveal whether 5-HT level is indeed involved in attentional allocation.

We thank the reviewer for this interesting suggestion. To test this idea, we compared the mouse's attentional indices and vertical/contrast energy sensitivities between two pairs of FAs: (1) FAs preceded by low DR-5HT occurring early in the block (after the cue switched sides) and all FAs; (2) FAs preceded by high DR-5HT occurring early in the block and all FAs. This comparison revealed weaker attention indices early in the block for both low and high DR-5HT (**Reviewer Fig. 8a**), consistent with prior work showing weaker attention following a block switch^{54,55,60}.

Interestingly, early-block attention indices were significantly higher for DR low as compared to DR high (**Reviewer Fig. 8a**). Importantly, these changes in DR-5HT activity and behavioral gain arose without effects on sensitivity to vertical and contrast energy (**Reviewer Fig. 8b-c**). Taken together, these data indicate that DR-5HT activity levels are selectively associated with the behavioral strength of attention (gain) rather than changes in the behavioral spatial sensitivity to visual features (focus). We now include this analysis in **Extended Data Figure 4** in the revised manuscript.

Reviewer Figure 8.a, Comparison of attention index computed from low or high DR-5HT signals preceding FAs occurring either soon after a block switch or to all FAs (n=6). Attention is weakest when DR-5HT is high during the early part of the block. **b**, Behavioral sensitivity to vertical or contrast energy computed from early or all FAs preceded by either high or low DR activity. **c**, Quantification of the spatial sensitivity maps shown in b. Euclidean distance was the summed absolute difference between early and all FAs. (n = 8, * is $p < 0.05$)

Rev2.2) The visual stimuli were lateralized in the task. Was the involvement of 5-HT in attentional modulation lateralized? Photometry and optogenetic stimulation in DR in the study appear to cover both hemispheres, however, 5-HT wide-field imaging in V1 was done unilaterally. Does the predictiveness of V1 5-HT in attention performance depend on laterality of the cue?

The reviewer asks if 5HT release is lateralized during our task. This issue is also included in our responses to **Reviewer 1**. To investigate lateralization, we performed new experiments where we used standard methods to virally express the genetically encoded 5HT sensor (GRAB5HT) across the entire mouse brain and then widefield imaged GRAB5HT signals from the entire dorsal cortex while mice attended and detected visual stimuli.

We first compared GRAB5HT dynamics in V1 before grating onset on cued-left and cued-right hit trials. This analysis shows that both V1s, regardless of cue location, experience the rapid, transient drop in 5HT that we causally linked to enhanced behavioral attention gain (**Reviewer Fig. 1b-e**). This preliminary result also suggests a potential neural mechanism for how changes in 5HT release might regulate attentional gain – by gating or modulating ongoing attentional computations that are already lateralized.

The circuitry behind attentional computations is poorly understood, but some work suggests that feedback inputs from midline frontal cortical areas (such as the anterior cingulate cortex) could play an important role^{14,26,61}. Considering these studies, it is interesting that both V1 and cingulate show similar dynamics before hits (**Reviewer Figs. 1d, 4b**) and false alarms (**Reviewer Fig. 4c**). It is also worth noting that 5HT release drops in a similar way in both V1s prior to hits (**Reviewer Fig. 1c**). More work will be needed to see if 5HT dynamics strengthen a lateralized ACC-V1 feedback during our task and whether such potential effects are related to visual attention.

Rev2.3) The authors appear to fixate that 5-HT modulation on visual attention is through V1. Indeed, authors found 5-HT dynamics in V1 resembled that of DR photometry recordings. However, such correlation was not causal. Optogenetic stimulation of DR can affect 5HT release in many other downstream targets. Besides V1, DR-5HT neurons also densely project to superior colliculus (SC) and exert potent suppression of SC visually evoked activity. Given SC is involved in visual attention, including in mice, it is plausible that 5-HT release in SC is playing an even more prominent role in attention. Ideally, optogenetic manipulation of 5-HT axon terminals in V1 could provide a definitive answer. The authors should at least discuss 5-HT release in other visual areas in attentional control.

We thank the reviewer for this important point and regret that our writing emphasized the idea that V1 is the sole site of action for 5HT during our task. Indeed, preliminary widefield imaging that we performed for this rebuttal suggest that the 5HT signals are present in both posterior visual

areas and midline frontal cortical areas (**Reviewer Figs 1d, 4b,c**). Optical stimulation of DR-5HT terminals is tricky since excitation could spread from the stimulated region into collaterals innervating other brain regions. We think a better experiment could be suppression of DR-5HT terminals in V1 and are in the process of validating various suppression tools including new opsins that specifically suppress neurotransmitter release⁶². Our revised manuscript now contains an expanded discussion of other cortical locations and the possibility of parallel action of 5HT.

Response to reviewer 3

Rev3.1) In the study “Rapid dynamics of dorsal raphe serotonin neurons selectively regulate the gain of visual attention” by Lehnert and colleagues, the authors show that fast dynamics of serotonin in dorsal raphe and V1 set the attentional gain, independent of location (using calcium imaging in DR, serotonin sensors in V1, and optogenetics). The paper is well written, the results are clear, and the conclusions are very important and well supported by the data. The discussion does a good job at linking these results with the broader literature on the topic.

We thank the reviewer for these very kind words and are delighted that they enjoyed the paper, felt the results were clear, and thought the conclusions important.

Rev3.2) I only have some minor points to raise, primarily for clarification purposes:

P2: “Gratings switched sides after ~35 trials and had a flat probability of appearance in time (Fig. 1b).” Fig 1b does not show that gratings switched nor the flat probability (I think this figure reference should be moved).

Corrected.

Rev3.3) What is the proportion of trials with FAs and misses?

FAs during the delay period do not terminate a trial in our task. Instead, such FAs just prolong the trial, until the delay period is lick-free; these FAs are used to compute behavioral attention measures. If the delay is lick-free, then a coherent grating is shown and a mouse can either hit or miss. Thus, the proportion of trials ending in a miss can be calculated from our psychometric curve of hits [% of misses = 100 - % of hits], which varies based on grating coherence.

When the three-bar grating finally appeared, approximately 14% of the time the grating coherence was zero and thus not visible. Because these other FAs occurred during the reaction time window, they allow us to estimate the proportion of hits that were due by chance, which is used for our d-prime calculations.

Rev3.4) Is the quantification in Figure 1g in the central bin only? Or average across bins? (Same for Figure 3i and 5k).

Our model fits spatial sensitivities and behavioral gain to cued and uncued checkerboards separately. The quantification in **Manuscript Fig. 1g** and elsewhere is the attentional index based on cued and uncued behavioral gain (model α_{cued} and α_{uncued}). The spatial organization of the sensitivity maps (β) are independent of α and not used in this measure. In **Manuscript Fig. 1e**, we show how sensitivity maps are scaled by behavioral gain.

Rev3.5) “These results reveal that a second-scale elevation in DR activity precedes a miss, whereas a rapid suppression precedes a hit.” Why do they show that there is an elevation? With respect to hits? Or with respect to baseline? If the former, then this should be rephrased to reflect what the data shows

It was indeed with respect to hits. We have rephrased this sentence.

Rev3.6) “These results showed that low DR activity predicts higher sensitivity to gratings rather than changes to impulsivity, engagement, or strategy” Can you clarify which results show which of these claims? Are RT and lapse rate associated with engagement? Response bias to strategy, and criterion to impulsivity?

We have clarified the use of these terms in the manuscript. We associate sensitivity to gratings with d-prime, hits, and threshold; impulsivity is measured as response bias; engagement is measured as lapse rate; and strategy measured as criterion.

Rev3.7) “Interestingly, sensitivity to vertical energy and local contrast remained at the center of the screen in both DR high and low (Fig. 3h). This result suggested that rapid DR low dynamics are correlated with increases in attentional gain rather than changes in the way sensitivity was allocated across the screen.” There are no stats to support this claim. Is the center-bias in DR high significant? (same applies to figure 4h and 4m)

The reviewer asks if the spatial sensitivities to vertical and contrast energy (β_v and β_c) are indeed the same between DR high and low, and between LED ON, LED off, and Ctrl. In **Reviewer Fig. 9** we compare model spatial sensitivity to vertical and contrast energy at the central binned checker to that of the surrounding bins for our photometry experiments (**Manuscript Fig. 3**). This comparison shows that model sensitivity remains centered, regardless of our photometry measures of DR-5HT activity (**Reviewer Fig. 9a-b**).

Reviewer Figure 9.a, Average model sensitivity maps for vertical and contrast energy computed from false alarms (FAs) preceded by high or low DR-5HT activity. **b**, Vertical and contrast sensitivities averaged for the center and all other surrounding checker bins for 8 individual mice during DR-5HT low and DR-5HT high. Sensitivity remains at the central bin regardless of DR-5HT activity. (n = 8 mice, ** is $p < 0.01$; *** is $p < 0.001$)

We also extended this analysis to model sensitivities obtained with optogenetic and GRAB5HT experiments. Model sensitivities remained centered in LED ON, LED OFF, ctrl, GRAB5HT high, and GRAB5HT low conditions. All these new analyses are integrated into the extended data figures of the revised manuscript and strengthen our claim that DR-5HT activity regulates the behavioral gain of visual attention rather than its spatial sensitivity (focus).

Rev3.8) Can you add some discussion to understand diffused effects of serotonin on the weighted sensitivity maps of 5i-j?

The reviewer wonders why the vertical energy attention maps for the GRAB5HT cohort appear more diffused than that of the photometry and optogenetic cohorts.

Briefly, it has to do with the nature of our task. We previously showed that mice monitor 3 main features when detecting the coherent 3-bar grating: Spatial frequency around 0.07 cycles per degree, vertical energy, and local contrast energy⁵⁹. In that study, we noticed variability in what mice chose to monitor. Many choose to monitor all 3 features, but some tend to emphasize one feature over another. Such biased mice often choose local contrast or vertical energy.

We are still trying to understand why some mice exhibit this bias, however, the GRAB5HT cohort contained some of these contrast-biased mice; the example mouse in **Manuscript Fig 5i** uses both. This is why the average vertical energy maps (**Manuscript Fig 5j**) appear more diffused whereas the contrast energy maps are more defined. We now include some discussion on this point and directly quantify the centered-ness of the sensitivity maps for GRAB5HT, photometry, and optogenetic cohorts through statistical comparisons located in the **Extended Data Figures 4-6**. New panels in **Extended Data Fig. 6c** show that 3 to 4 mice chose contrast over vertical energy which is why the average looks more diffuse for this feature.

Response to reviewer 4

Rev4.1) In this manuscript, Lehnert et al. show that 5-HT modulates visual attention, with decreases in 5-HT neural activity and release in V1 resulting in greater attention towards visual cues. The authors employ a useful behavioral paradigm for tracking attention in addition to fiber photometry, optogenetics and widefield imaging to track and manipulate DR neurons/5-HT release during execution of this task. Consistent findings across these techniques are a strength of the manuscript. The data could be more clearly presented and described.

We thank the reviewer for their time in reading the manuscript. In response to this and the other reviewers, we have made numerous changes to improve the clarity of the manuscript.

Rev4.2) Showing fluorescence data without filters applied would be a useful supp figure given the authors noted caveat the filters can change activity around large deviations (even if the authors removed the Hit activity prior to the filter, it would be good to show less processed data that show this to be the case)

We thank the reviewer for this suggestion. In **Reviewer Fig. 10**, we show our raw 470 nm and 405 nm fluorescent signals from an entire session of one example mouse, the unfiltered signal following 405 correction, the same signal following high-pass filtering, and the DR-5HT signal obtained from the various pre-processing steps. We now include these panels in **Extended Data Fig. 1** in the revised manuscript.

Reviewer Figure 10. **a**, Example session fluorescence signal evoked by 470 nm and 405 nm LEDs during photometry. **b**, The same 470 nm signal as in panel a following regression to the 405 nm signal. **c**, a high-pass filtered version of the signal in **b**. **d**, magnified view of the blue boxed region of the unfiltered and filtered signals in **c**. **e**, example photometry signals taken from the indicated processing stages illustrated in **a-d**.

Rev4.3) Are there differences in behavior and/or 5-HT activity in trials that follow a FA vs trial start lick?

In trained mice, all trials start with a lick (start lick, **Manuscript Fig. 2f**) and are followed by a delay period. We have very few delay periods that are false-alarm free. Most mice false alarmed to noise, which we previously explained as occurring because the noise patterns happen to look like the stimulus⁵⁹. Most of our FAs (>90%) follow other FAs rather than following a start lick. Our label “start or FA licks” in **Manuscript Figs. 2g&5d** may have misled the reviewer. We have revised these labels to “FA licks”.

Rev4.4) Figure 3 d prime and threshold plots should start at 0 on the y axis.

We used the y-scaling in order to make it easier to see our differences in d-prime and threshold. The same is true for reaction time which begins at 0.4 and ends at 0.7. We respectfully feel this is the best way to report our differences which are significant for d-prime and threshold, and so have left these axes labels the same. We can change this, however, if the reviewer and editor feel strongly about this issue.

Minor comments:

Rev4.5) -5HT recording data at the level of V1 does not correlate as robustly with attention as photometry data, and gain/loss of function data is at the level of the DR. This limits the ability to state that the 5-HT modulation is at the level of V1 (or at least only at the level of V1).

The reviewer points out that V1 may not be the only site at which rapid DR-5HT low signals act to regulate the gain of visual attention. We agree, and based on comments from Reviewers 1&2, we have now softened this emphasis on V1 as the sole site of 5HT action throughout the manuscript and have included a new discussion paragraph that summarizes all candidate cortical areas.

Rev4.6) -In some supp figure legends delta f/F is written as delta f/Ff

Corrected.

-Colors in supplemental figures often re-use blue/orange color scheme despite not referring to the same features as the main figures (e.g., Figure S3 uses blue/orange for hit/miss and has entirely new colors for high/low DR activity). Figures should have consistent themes to help readability.

We have changed the color scheme of hits and misses in **Extended Data Figure 3** in the revised manuscript. We retain the green/purple DR high and low in this figure to distinguish these “slow” DR high and low states from the fast ones in **Manuscript Figs 2,3,5** that we have associated with the gain of visual attention.

References

1. Luo, M., Li, Y. & Zhong, W. Do dorsal raphe 5-HT neurons encode ‘beneficialness’? *Neurobiol Learn Mem* **135**, 40–49 (2016).
2. Cohen, J. Y., Amoroso, M. W. & Uchida, N. Serotonergic neurons signal reward and punishment on multiple timescales. *Elife* **4**, (2015).
3. Ren, J. *et al.* Anatomically Defined and Functionally Distinct Dorsal Raphe Serotonin Subsystems. *Cell* **175**, 472-487.e20 (2018).
4. Miyazaki, K. W. *et al.* Optogenetic activation of dorsal raphe serotonin neurons enhances patience for future rewards. *Current biology: CB* **24**, 2033–2040 (2014).

5. Grossman, C. D. & Cohen, J. Y. Neuromodulation and Neurophysiology on the Timescale of Learning and Decision-Making. *Annu Rev Neurosci* **45**, 317–337 (2022).
6. Grossman, C. D., Bari, B. A. & Cohen, J. Y. Serotonin neurons modulate learning rate through uncertainty. *Current Biology* **32**, 586-599.e7 (2022).
7. Muzerelle, A., Scotto-Lomassese, S., Bernard, J. F., Soiza-Reilly, M. & Gaspar, P. Conditional anterograde tracing reveals distinct targeting of individual serotonin cell groups (B5-B9) to the forebrain and brainstem. *Brain Struct Funct* **221**, 535–561 (2016).
8. Chebolu, S., Dayan, P. & Lloyd, K. Vigilance, arousal, and acetylcholine: Optimal control of attention in a simple detection task. *PLoS Comput Biol* **18**, e1010642 (2022).
9. Gritton, H. J. *et al.* Cortical cholinergic signaling controls the detection of cues. *Proc Natl Acad Sci U S A* **113**, E1089-1097 (2016).
10. Pinto, L. *et al.* Fast modulation of visual perception by basal forebrain cholinergic neurons. *Nat Neurosci* **16**, 1857–1863 (2013).
11. Sarter, M. & Lustig, C. Cholinergic double duty: cue detection and attentional control. *Curr Opin Psychol* **29**, 102–107 (2019).
12. Parikh, V. & Bangasser, D. A. Cholinergic Signaling Dynamics and Cognitive Control of Attention. *Curr Top Behav Neurosci* **45**, 71–87 (2020).
13. Noudoost, B. & Moore, T. The role of neuromodulators in selective attention. *Trends Cogn Sci* **15**, 585–591 (2011).
14. Huda, R. *et al.* Distinct prefrontal top-down circuits differentially modulate sensorimotor behavior. *Nat Commun* **11**, 6007 (2020).
15. Okaty, B. W. *et al.* A single-cell transcriptomic and anatomic atlas of mouse dorsal raphe Pet1 neurons. *Elife* **9**, e55523 (2020).
16. Muzerelle, A., Scotto-Lomassese, S., Bernard, J. F., Soiza-Reilly, M. & Gaspar, P. Conditional anterograde tracing reveals distinct targeting of individual serotonin cell groups (B5-B9) to the forebrain and brainstem. *Brain Struct Funct* **221**, 535–561 (2016).
17. Weber, E. T. & Andrade, R. Htr2a Gene and 5-HT(2A) Receptor Expression in the Cerebral Cortex Studied Using Genetically Modified Mice. *Front Neurosci* **4**, 36 (2010).

18. Okaty, B. W., Commons, K. G. & Dymecki, S. M. Embracing diversity in the 5-HT neuronal system. *Nat Rev Neurosci* **20**, 397–424 (2019).
19. Lambe, E. K., Fillman, S. G., Webster, M. J. & Shannon Weickert, C. Serotonin receptor expression in human prefrontal cortex: balancing excitation and inhibition across postnatal development. *PLoS One* **6**, e22799 (2011).
20. Salvan, P. *et al.* Serotonin regulation of behavior via large-scale neuromodulation of serotonin receptor networks. *Nat Neurosci* **26**, 53–63 (2023).
21. Berger, M., Gray, J. A. & Roth, B. L. The expanded biology of serotonin. *Annu Rev Med* **60**, 355–366 (2009).
22. Couto, J. *et al.* Chronic, cortex-wide imaging of specific cell populations during behavior. *Nat Protoc* **16**, 3241–3263 (2021).
23. Chan, K. Y. *et al.* Engineered AAVs for efficient noninvasive gene delivery to the central and peripheral nervous systems. *Nat Neurosci* **20**, 1172–1179 (2017).
24. Wan, J. *et al.* A genetically encoded sensor for measuring serotonin dynamics. *Nat Neurosci* **24**, 746–752 (2021).
25. Feng, J. *et al.* Monitoring norepinephrine release *in vivo* using next-generation GRAB_{NE} sensors. Preprint at <https://doi.org/10.1101/2023.06.22.546075> (2023).
26. Zhang, S. *et al.* Selective attention. Long-range and local circuits for top-down modulation of visual cortex processing. *Science* **345**, 660–665 (2014).
27. Huang, K. W. *et al.* Molecular and anatomical organization of the dorsal raphe nucleus. *Elife* **8**, e46464 (2019).
28. Cho, J. R. *et al.* Dorsal Raphe Dopamine Neurons Signal Motivational Salience Dependent on Internal State, Expectation, and Behavioral Context. *J Neurosci* **41**, 2645–2655 (2021).
29. Li, Y. *et al.* Serotonin neurons in the dorsal raphe nucleus encode reward signals. *Nat Commun* **7**, 10503 (2016).
30. Hong, S. Z. *et al.* Norepinephrine potentiates and serotonin depresses visual cortical responses by transforming eligibility traces. *Nat Commun* **13**, 3202 (2022).

31. Azimi, Z. *et al.* Separable gain control of ongoing and evoked activity in the visual cortex by serotonergic input. *Elife* **9**, e53552 (2020).
32. Seillier, L. *et al.* Serotonin Decreases the Gain of Visual Responses in Awake Macaque V1. *J Neurosci* **37**, 11390–11405 (2017).
33. Patel, A. M., Kawaguchi, K., Seillier, L. & Nienborg, H. Serotonergic modulation of local network processing in V1 mirrors previously reported signatures of local network modulation by spatial attention. *Eur J Neurosci* **57**, 1368–1382 (2023).
34. Barzan, R. *et al.* Gain control of sensory input across polysynaptic circuitries in mouse visual cortex by a single G protein-coupled receptor type (5-HT_{2A}). *Nat Commun* **15**, 8078 (2024).
35. Reggiani, J. D. S. *et al.* Brainstem serotonin neurons selectively gate retinal information flow to thalamus. *Neuron* **111**, 711-726.e11 (2023).
36. Chen, C. & Regehr, W. G. Presynaptic modulation of the retinogeniculate synapse. *J Neurosci* **23**, 3130–3135 (2003).
37. Seeburg, D. P., Liu, X. & Chen, C. Frequency-dependent modulation of retinogeniculate transmission by serotonin. *J Neurosci* **24**, 10950–10962 (2004).
38. Sherman, S. M. & Guillery, R. W. *Functional Connections of Cortical Areas: A New View from the Thalamus.* (The MIT Press, London, England ; Cambridge, Mass., 2013).
39. Kayama, Y., Shimada, S., Hishikawa, Y. & Ogawa, T. Effects of stimulating the dorsal raphe nucleus of the rat on neuronal activity in the dorsal lateral geniculate nucleus. *Brain Res* **489**, 1–11 (1989).
40. Michaiel, A. M., Parker, P. R. L. & Niell, C. M. A Hallucinogenic Serotonin-2A Receptor Agonist Reduces Visual Response Gain and Alters Temporal Dynamics in Mouse V1. *Cell Rep* **26**, 3475-3483.e4 (2019).
41. Syeda, A. *et al.* *Facemap: A Framework for Modeling Neural Activity Based on Orofacial Tracking.* <http://biorxiv.org/lookup/doi/10.1101/2022.11.03.515121> (2022)
doi:10.1101/2022.11.03.515121.

42. Carrasco, M., Penpeci-Talgar, C. & Eckstein, M. Spatial covert attention increases contrast sensitivity across the CSF: support for signal enhancement. *Vision Res* **40**, 1203–1215 (2000).
43. Fernández, A., Okun, S. & Carrasco, M. Differential Effects of Endogenous and Exogenous Attention on Sensory Tuning. *J Neurosci* **42**, 1316–1327 (2022).
44. Giordano, A. M., McElree, B. & Carrasco, M. On the automaticity and flexibility of covert attention: a speed-accuracy trade-off analysis. *J Vis* **9**, 30.1–10 (2009).
45. Kirsch, W., Heitling, B. & Kunde, W. Changes in the size of attentional focus modulate the apparent object's size. *Vision Research* **153**, 82–90 (2018).
46. Liu, T. & Mance, I. Constant spread of feature-based attention across the visual field. *Vision Research* **51**, 26–33 (2011).
47. Müller, M. M. *et al.* Feature-selective attention enhances color signals in early visual areas of the human brain. *Proc. Natl. Acad. Sci. U.S.A.* **103**, 14250–14254 (2006).
48. Mitchell, K. J. The genetics of neurodevelopmental disease. *Curr Opin Neurobiol* **21**, 197–203 (2011).
49. Cohen, M. R. & Maunsell, J. H. R. Attention improves performance primarily by reducing interneuronal correlations. *Nat Neurosci* **12**, 1594–1600 (2009).
50. Ruff, D. A. & Cohen, M. R. Simultaneous multi-area recordings suggest that attention improves performance by reshaping stimulus representations. *Nat Neurosci* **22**, 1669–1676 (2019).
51. Ruff, D. A., Xue, C., Kramer, L. E., Baqai, F. & Cohen, M. R. Low rank mechanisms underlying flexible visual representations. *Proc Natl Acad Sci U S A* **117**, 29321–29329 (2020).
52. Cook, E. P. & Maunsell, J. H. R. Attentional modulation of motion integration of individual neurons in the middle temporal visual area. *J Neurosci* **24**, 7964–7977 (2004).
53. Cohen, M. R. & Maunsell, J. H. R. A neuronal population measure of attention predicts behavioral performance on individual trials. *J Neurosci* **30**, 15241–15253 (2010).

54. Speed, A., Del Rosario, J., Mikail, N. & Haider, B. Spatial attention enhances network, cellular and subthreshold responses in mouse visual cortex. *Nat Commun* **11**, 505 (2020).
55. Kanamori, T. & Mrsic-Flogel, T. D. Independent response modulation of visual cortical neurons by attentional and behavioral states. *Neuron* **110**, 3907-3918.e6 (2022).
56. McBride, E. G., Lee, S.-Y. J. & Callaway, E. M. Local and Global Influences of Visual Spatial Selection and Locomotion in Mouse Primary Visual Cortex. *Curr Biol* **29**, 1592-1605.e5 (2019).
57. Myers-Joseph, D., Wilmes, K. A., Fernandez-Otero, M., Clopath, C. & Khan, A. G. Disinhibition by VIP interneurons is orthogonal to cross-modal attentional modulation in primary visual cortex. *Neuron* **112**, 628-645.e7 (2024).
58. Poort, J. *et al.* Learning and attention increase visual response selectivity through distinct mechanisms. *Neuron* **110**, 686-697.e6 (2022).
59. Lehnert, J. *et al.* Visual attention to features and space in mice using reverse correlation. *Curr Biol* **33**, 3690-3701.e4 (2023).
60. Speed, A. & Haider, B. Probing mechanisms of visual spatial attention in mice. *Trends Neurosci* **44**, 822–836 (2021).
61. Zhang, S. *et al.* Long-range and local circuits for top-down modulation of visual cortex processing. *Science* **345**, 660–665 (2014).
62. Mahn, M. *et al.* High-efficiency optogenetic silencing with soma-targeted anion-conducting channelrhodopsins. *Nat Commun* **9**, 4125 (2018).

We provide a point-by-point response to reviewer comments below and indicate where new data/text has been added to the manuscript.

Reviewer #1 (Remarks to the Author): I am grateful to the authors for their detailed and thoughtful replies to my initial questions. Despite these clarifications, I still find it difficult to fully grasp the impact and broader meaning of the results presented in this paper. My second round of questions is therefore aimed at achieving a more precise understanding.

We thank the reviewer for acknowledging our prior revisions. The reviewer requests a more precise understanding of the meaning and impact of our study.

Meaning: Our central finding is that rapid DR-5HT dynamics specifically regulates how strongly a mouse attends to task-relevant features on the cued screen versus those on the uncued screen. We used a behavioral method (Manuscript Fig. 1) to measure this quantity (hereafter termed **attentional strength**) and identify where on the screen they attended for task-relevant features (hereafter termed **attentional focus**). We saw that rapid endogenous and optogenetically-evoked DR-5HT dynamics selectively regulated attentional strength but did not affect attentional focus or measures of states (eg: impulsivity, arousal) unrelated to the stimulus. This result suggests that (1) the brain controls attentional strength and focus separately; and (2) that rapid DR-5HT neuromodulation regulates attentional strength.

Impact: These findings are impactful for several reasons: (1) It is the first demonstration that DR and 5HT regulate visual attention; (2) It is the first demonstration that DR-5HT specifically controls attentional strength; (3) It reveals that attentional strength and focus are computationally separable phenomena; and (4) It is the first report of perceptually-relevant, second-scale dynamics in DR activity and 5HT release. While these points alone make our study impactful to neuroscientists interested in visual circuits, 5HT biology, and neuromodulation, we now also formalize our results in a computational model which explains our principal findings using the canonical cortical computation of normalization. This new aspect will also make our study impactful to theorists and experimentalists working on cortical computation.

Rev1a Scope of the main findings

The primary result of this paper is not a modulation of visual processing by serotonin within visual cortex (as I originally suspected in Question 1), but rather a more general effect of serotonin on visual detection behavior.

The reviewer is correct that our paper is not about 5HT modulation of visual cortical (V1) neurons. However, we respectfully point out that this was not a claim we made nor was it a central goal of our revision or original submission.

Moreover, we respectfully disagree with the reviewer's statement that our findings show an unspecific general effect of serotonin on visual detection behavior. Had DR-5HT non-specifically modulated sensitivity to stimulus energy (ie: a general effect on visual detection) it would have impacted both cued and uncued sides equally and would not change attentional strength (since it reflects cued-uncued difference). Instead, our studies show that rapid DR-5HT dynamics specifically regulate attentional strength. We arrive at this conclusion because:

1. DR-5HT fluctuations before false alarms (FAs) predicted attentional strength: Low DR-5HT predicted strong attention whereas high DR-5HT predicted weak or no attention (**Manuscript Fig. 3,5**). We measured attentional strength and focus from FAs (**Manuscript Fig. 1**)
2. Rapid DR-5HT dynamics also predicted grating detection (performance) which was likely due to its effects on attentional strength (**Manuscript Fig. 2-5**).
3. DR-5HT fluctuations before FAs did not affect the locations within a screen a mouse chose to attend (attentional focus), nor did they affect FA rate (**Manuscript Ext. Data Fig. 4,5**)
4. DR-5HT fluctuations before grating onset did not affect guess rate (impulsivity), lapse rate (motivation), or criterion (strategy) (**Manuscript Fig. 3-5**).
5. Only rapid DR-5HT fluctuations predict changes in attentional strength (**Manuscript Ext. Data Fig. 3**); we see slow fluctuations, like others do, but they were unrelated to our behavioral measures.
6. Optogenetic DR excitation before FAs reduced the attentional strength to zero (**Manuscript Fig. 4**,

also see **Rev.2** below); excitation before grating onset reduced performance, likely due to low attentional strength.

7. Optogenetic DR suppression before FAs increased attention strength (**Manuscript Fig. 4**); suppression before grating onset improved performance, likely due to high attentional strength.
8. Neither optogenetic perturbation changed our measures of attentional focus, impulsivity, motivation, or strategy (**Manuscript Fig. 4, Ext. Data Fig. 5**).

We have revised text throughout the introduction and results to make this clearer. We have also substantially revised our discussion using a computational model (see below).

Rev1b The authors' data do not support any claim of a gain change in V1, even if such a mechanism remains possible. Indeed, as they note in the first paragraph of the discussion section ("How does DR-5HT contribute to attentional network"), serotonin's action on V1 networks could contribute to attentional gain modulation. Given that serotonin is generally believed to suppress cortical excitatory neurons, a decrease in serotonin release within V1 would be expected to increase neuronal excitability.

The reviewer is correct that our data does not support any claim of gain change in V1. However, as we mention above, this was not a claim we made. Our past **Manuscript Discussion** used prior work to speculate that 5HT's suppression of visual neurons could be how it regulates attentional strength. Here we address this prediction and others by formalizing our results in a computational model that captures our key findings: that non-lateralized DR-5HT specifically and inversely regulates attentional strength.

We obtained and adapted the code for the highly-cited and influential normalization model of attention¹ (**Rev. Fig. 1a**). We chose this particular model because normalization is a canonical neural computation², a well-accepted feature of visual cortical circuits¹, and because this model accounts for many observed behavioral and neural attentional signatures (eg: contrast/response/feature-similarity gain, biased competition)¹⁻⁵.

Briefly, left and right stimuli drive brain representations that are multiplied by internal attention fields for cued and uncued sides. In this way, attention fields selectively boost one representation without affecting the other (**Rev. Fig.1a**). Cued and uncued representations are then pooled over features and space to create suppressive drive. Finally, cued and uncued representations are divided by the suppressive drive (normalized) to produce a stronger population response on the cued side (*right stimulus is cued*). Since prior work showed this neural population model also explains behavioral effects of attention¹⁻⁵, we interpret differences in right versus left model population responses (model attentional strength) as producing our behavioral measure of attentional strength.

We hypothesized that DR-5HT scaled the suppressive drive, a key computation regulating model attentional strength. To test this, we multiplied the suppressive drive by a single parameter reflecting DR-5HT and observed the model's predictions (**Rev. Fig.1b-c**).

Model output aligned exactly with our observations. Model population responses were stronger on the cued side than the uncued side when DR-5HT was low (**Rev. Fig.1b**). This is because as the denominator (suppressive drive) shrinks, the attention field's effects on model output grow. Interestingly, cued and uncued population responses were roughly equal when DR-5HT is high (**Rev. Fig.1c**). This is because as the denominator (suppressive drive) grows, the attention field's effects shrink. Computing an index from the maximum population response to cued and uncued confirms these observations and shows higher model attention index when DR-5HT is low versus when it is high (**Rev. Fig.1d**). Note that changes in DR-5HT do not affect the attention field itself (model attentional focus), rather DR-5HT only scales the attention field's effect on population responses (model attentional strength). Thus, our normalization model suggests a computational mechanism for separating attentional strength and focus (**Rev. Fig.1e**).

It is important to point out that the model could have failed to capture our results, regardless of where we put DR-5HT in its architecture. However, DR-5HT specifically mimicked our results when it scaled the suppressive drive. Since our model inherits the original model's biologically-rooted architecture (eg: stimulus representations compete, top-down attention field, suppressive normalization, etc), it predicts that 5HT should suppress cortical neurons holding representations of attention targets. This prediction is

Reviewer Figure 1. Normalization model of attention and DR-5HT. **a**, Simulation of the normalization model of attention from Reynolds and Heeger 2009. Stimulus drive represents the neural population response to left and right checkerboard stimuli if there were no attention or normalization. Stimulus drive is depicted with receptive field location along the horizontal and feature preference (orientation, etc) along the vertical. The attention field depicts the effect of cueing the visual location of the right stimulus but is assumed to be non-specific to visual features. The attention field multiplies the stimulus drive, and its product is pooled over space and features to generate the suppressive drive. The panel on the right shows the population response (R) of neurons produced by dividing the product of stimulus drive and attention field by the suppressive drive. **b**, Modified normalization model includes a term to scale the suppressive field by DR-5HT (DR-5HT drive). Low DR-5HT strengthens attention to the cued side. **c**, Same as b, but for DR-5HT high which weakens attention to the cued side. Maximum population response to cued and uncued indicated ($R_{\text{max_cued}}$ and $R_{\text{max_uncued}}$). **d**, Top: equation to computed model attention index from $R_{\text{max_cued}}$ and $R_{\text{max_uncued}}$; Bottom: Model attention index computed from population responses shown in b (low DR-5HT drive) and c (high DR-5HT drive). Higher model attention index when DR-5HT is low versus when it is high mimics our behavioral results showing increased attentional strength. **e**, Cartoon of modified normalization model with attentional focus and attentional strength indicated. The attention field focuses the circuit on target features and DR-5HT signals scale suppressive drive to regulate how strongly such targets are amplified in the population responses that underlie behavioral attention.

supported by reports of excitatory 5HT receptors expression in cortical interneurons⁶⁻¹⁰, inhibitory 5HT receptors expression in pyramidal neurons¹¹⁻¹³, and optogenetic evidence showing suppression of V1 pyramidal neurons by DR axon stimulation^{14,15}.

The model also shows how non-lateralized 5HT produces lateralized enhancement of cued output (see our response to **Rev. 2** below) and explains how DR-5HT inversely controls attentional strength (see our response to **Rev. 3c** below). Since our modified normalization model provides a useful scope for our results and discussion, we now include it as **Manuscript Fig 6** and describe it in the methods section. We also further de-emphasize 5HT's effects in V1 in our introduction but retain it in the discussion.

Rev.2. On lateralization

My second concern stemmed from the apparent lateralization of attention to the visual stimulus, which initially raised the possibility that serotonin might contribute to lateralized visual processing. However, the authors' preliminary results do not support this view: no evidence of lateralization in serotonin-related activity was observed. The source of this lateralization therefore remains unknown and is unlikely to be directly explained by lateralized serotonergic modulation. As the authors themselves note in their reply to Question 2, serotonin is more likely involved in setting a general attentive brain state, albeit in a negative fashion (see comment 3 below). In this context, the observation that excitation of DR-5HT neurons is associated with a switch of attention to the unattended side (Fig. 4i) is difficult to reconcile with their

interpretation. This finding is neither highlighted in the results section nor addressed in the discussion.

There are 3 points here, the reviewer: 1) suggests a non-lateralized 5HT signal cannot produce lateralized attentional effects; 2) suggests that **Manuscript Fig. 4i** supports “a switch of attention to the unattended side”; 3) suggests our 5HT results set a “*general attentive brain state*”.

1. Enhanced attention to the cued side can arise from non-lateralized DR-5HT signals if 5HT regulates normalization. The reviewer hypothesizes that only a lateralized 5HT signal could underlie lateralized attention to the cued side and states that our preliminary results do not support their hypothesis. However, we hypothesized in our past response that DR sends a global (ie: non-lateralized) signal to enhance ongoing lateralized mechanisms that embody attentional focus.

Our modified normalization model, which uses a non-lateralized (ie: global) multiplicative term for DR-5HT, refines the hypothesis in our prior response: non-lateralized DR-5HT regulates the suppressive field, and through normalization, scales the effect of the attention field on the population response (**Rev. Fig. 1b-d**). Thus, DR-5HT regulates how strongly the model attends the cued side but does not affect what attention focuses upon (**Rev. Fig. 1e**). These model results are remarkably similar to our main findings.

2. We interpret Fig 4i as “no attention” rather than a switch to the uncued side: We thank the reviewer for pointing this out to us and we can see how it may appear to indicate a switch of attention to the unattended side because we did not include statistics to show otherwise. However, the effect across all mice is not significantly different from zero (ie: not attending to either side) and was influenced by two mice that happened to show unusually strong uncued model gain. The collective results shown in **Manuscript Figs. 1g, 3i, 4n, 5k, and Extended Fig. 4c**, are consistent with each other and support our claim that when DR-5HT is high, attention to the cued screen weakens. For these reasons, we interpret **panel 4i** as “no attention” rather than a switch to the uncued side.

We now indicate that this point is not significantly different from zero in the figure legend and results. We also discuss how high DR-5HT leads to “weak attention” using our modified normalization model in the **revised discussion**.

3. DR-5HT does not set “a general attentive brain state”, it regulates the strength of attention. We respectfully direct the reviewer to our response to this point above in **Rev.1a**.

Rev.3a. Interpretation of DR-5HT suppression

....The authors present the suppression of DR-5HT activity during successful sensory processing as evidence for an active involvement in attention.

The reviewer suggests that we used DR-5HT low dynamics “during” successful sensory processing (we assume this means DR-5HT when a mouse detects the grating or hits) as evidence that DR-5HT was involved in attention. We apologize that this was not clearer, but while the relationship between pre-grating DR-5HT suppression and hits is consistent with attention, it is not our main evidence for this point. The direct evidence comes from comparing behavioral attention measures between false alarm licks preceded by high versus low DR-5HT activity. We drew a causal (active) link between DR-5HT and attentional strength with optogenetics (**Manuscript Fig. 4**): When we optogenetically increased DR activity before false alarms, we reduced attentional strength; optogenetically decreasing DR activity increased attentional strength.

Rev.3b. Yet, DR-5HT neurons are less active during these hit trials. Their activity instead increases during reward processing.

The reviewer suggests that: (1) DR-5HT neurons are less active on hit trials; (2) the rapid suppression that controls attention is actually due to post-grating reward transients.

(1) Rapid DR-5HT dynamics fluctuate across both hit and miss trials as animals perform the task. Average DR-5HT tends to be low just before grating onset on trials that mice correctly detect (**Manuscript Fig. 2h**). During the trial itself, DR-5HT activity is not significantly different for hit and miss trials (**Rev.**

Fig. 2a-c). We think our poor labelling of **Manuscript Fig. 2h** could lead to the interpretation that it shows the entire trial length when it does not. Our trial lengths are variable (secs to minutes), because delay period FAs seamlessly restart the cue. Both long and short trials can end in hit or miss, and all trials exhibit rapid DR-5HT dynamics throughout their entire length (**Rev. Fig. 2a**, *new Manuscript Fig. 2f*).

Manuscript Figure 2h *Hit only* shows the final portion of many hit trials sorted by delay length; there were many FAs and trial restarts before this final portion. **Manuscript Figure 2i** compares mean $\Delta F/F$ for this final part of hit and miss trials and shows this mean is lower for hits. We have clarified the labelling to indicate that only the last 15sec of the trial are shown in **Manuscript Fig. 2h**.

When the entire pre-grating trial length is considered, signal distributions of DR-5HT for hits and misses are indistinguishable (an example mouse shown in **Rev. Fig. 2b**). Repeating this analysis across mice showed no significant difference in the mean of such hit and miss distributions (**Rev. Fig. 2c**). This is consistent with rapid DR-5HT dynamics which fluctuate many times over the length of a trial.

All the rapid DR-5HT dynamics we analyzed happened on timescales faster than our overall average trial length of 34sec (**Rev. Fig. 2d**). On the other hand, slow dynamics did not impact behavior in our task (**Manuscript Ext. Data Fig. 3**). Example dynamics are shown in **Reviewer Fig. 2a**, where many periods of high, low, and baseline DR-5HT signals appear within a trial. Taken together, these results show that the entire trial's DR-5HT activity before grating onset is similar for hit and miss trials.

(2) Pre-grating suppression does not depend on post-grating reward transient. We do not think the post-grating reward transient explains the rapid pre-grating dynamics because (1) DR-5HT rapidly fluctuates between the reward transient and the next pre-grating period (**Rev. Fig. 2a**); (2) We directly

REDACTED

suppressed or elevated DR-5HT with optogenetics which significantly increased or decreased attention to the cued side.

As further proof, we saw no significant correlation between (1) the reward transient on hit trials and the next trial's pre-grating $\Delta F/F$ (**Rev. Fig. 2e**); (2) the pre- and post-grating $\Delta F/F$ for all hit trials (**Rev. Fig. 2f**). This latter correlation is flat because it uses hits at all coherences (recall high coherence hits are not predicted by pre-grating DR-5HT) and because the trial-level reward transients are variable (**Manuscript Fig. 3b-c**), the overall correlation is flat. Taken together, these new results and those in our manuscript demonstrate that rapid DR-5HT fluctuations (that regulate attention) and post-grating transients (which prior work linked to reward) are separate phenomena. A reasonable hypothesis for the differing roles for these two signals is that they come from different DR cell types whose signals are combined when reading from the optical fiber. We have integrated **Reviewer Fig. 2e-f** into **Manuscript Extended Data Fig. 2**.

Rev.3c. A central aspect of the narrative strikes me as counterintuitive across the manuscript....The paper frames this suppression as an "active implication," which risks inverting the logic of interpretation. To illustrate: it is as if a newspaper were to write, "Traffic accidents regulate improvements in city traffic," when in reality accidents hinder traffic flow rather than drive its improvement.

We apologize but could not understand the reviewer's analogy and hope the reviewer forgives any resulting misinterpretation of their request.

We think the reviewer is asking if (1) DR suppression or reward transient actively regulates visual attention and/or (2) whether inverted control of attention by DR-5HT is a reasonable biological mechanism.

1) Rapid DR-5HT dynamics that control attention are separate from the post-grating reward transient. It is possible the reviewer thinks that only post-grating reward-associated increase in DR-5HT can be considered an active process and it is counterintuitive to think that DR-5HT fluctuations outside of this increase could be considered active. We strongly disagree with this premise.

First, we show that the ongoing DR-5HT dynamics (high-low) are rapid and unrelated to the size of the reward transient (**Rev. Fig. 2e-f**). This means that DR-5HT dynamics (that regulate attention) are separate from the post-grating responses (associated with reward). Second, the dynamic value of DR-5HT is causally linked to attentional strength –our rapid optogenetic manipulations (**Manuscript Fig. 4**) mimic our photometry and GRAB5HT results (**Manuscript Fig.3,5**). Given these points, we feel it is reasonable to conclude that rapid DR-5HT fluctuations actively regulate visual attention.

(2) Inverse control is a canonical feature of many neural computations. Inverse relationships characterize many important brain signals. For example, the output neurons of the basal ganglia tonically inhibit thalamic neurons involved in movement; but become suppressed to release these thalamic neurons and cause movement. There are many other examples, including many stages of the visual system (eg: photons and photoreceptor potentials; OFF responses), or the many examples of disinhibition which are cardinal features of many neural computations and circuits in the brain.

Our modified normalization model also supports an inverse relationship between attention and DR-5HT because it is part of the model's suppression. High DR-5HT strengthens the suppressive field which weakens attention to the cued side; Low DR-5HT weakens the suppressive field which strengthens attention to the cued side. This suppressive role for DR-5HT, as the reviewer acknowledges, is supported by prior work on 5HT receptor expression within cortical types and optogenetic studies^{6–15}. We should also note that DR neurons are tonically active and the suppression of DR neuron activity by stimuli has been reported before¹⁶.

A final (speculative) point is that the inverse relationship between DR-5HT and attentional strength may mean another signal with a non-inverted relationship balances DR-5HT's effects, creating push-pull dynamics that could allow rapid attentional control. Candidates for such a signal could be acetylcholine^{17–22} and/or norepinephrine^{23–26}. Studying these neuromodulators with our task could be a way to address

this issue. We have integrated this speculation into our **revised Manuscript Discussion**.

Rev.3d. Taken together, the key finding is that DR-5HT activity is suppressed immediately before hit trials in a visual detection task.

The reviewer suggests that our key finding is the “pre-grating DR-5HT suppression on hit trials”

We apologize that this was not clearer, as this is not the key finding of our work. Our key finding is that rapid DR-5HT dynamics (high and low) causally and selectively regulated attentional strength. The pre-grating suppression before hits is a subset of the data in support of this finding –the direct evidence comes from studying our behavioral attention measures in response to endogenously or optogenetically-induced changes in DR-5HT: attentional strength increases when DR-5HT is low before FAs and attentional strength decreases when DR-5HT is high.

Rev.3e. Yet, without a clear neuronal mechanism linking this activity to behavior, the broader relevance of the study for the neuroscience community remains limited.

While we appreciate the reviewer’s enthusiasm for a 5HT mechanism in V1, we strongly disagree that it is required for our study to be impactful. This is because we report several novel observations that many neuroscience communities will find important. They are:

1) The first report that DR and 5HT regulate visual attention. This holds immediate impact for attention research because DR-5HT dynamics could be used as a real-time readout of when mice deploy attention. For cognitive neuroscientists, it motivates study into whether 5HT-modifying drugs may ameliorate attentional deficits that accompany disorders like ADHD and autism spectrum disorder. Thus, our findings offer new entry points to understand attentional circuitry and investigate therapeutic potential of 5HT signaling for attentional dysfunction.

2) Demonstration that DR-5HT specifically regulates attentional strength. This neuronal mechanism guides circuit dissection studies to learn how 5HT influences attentional target representations whose distributed representations across frontal, parietal, and visual cortical areas match the broad DR innervation of cortex. This observation also links the fields of neuromodulation, attention, and visual neuroscience and offers our task/behavioral methods to place other neuromodulators into a common framework that includes attention, internal state (arousal, motivation), and perceptually guided decisions. Thus, our study offers a strategy to unify other candidate attention signals into a single picture and guides studies into their circuit-level mechanisms.

3) Reveals that attentional strength and focus are computationally separable. This insight would interest visual and computational efforts to understand the architecture of attention computations. Our data suggests that 5HT encodes suppressive field strength in the normalization model of attention, but our data is also consistent with 5HT encoding cue validity as captured by reinforcement learning models. Thus, our findings suggest a way to bridge theoretical models of attention and learning.

4) Discovery of perceptually-relevant, second-scale, DR-5HT dynamics would interest DR and 5HT biologists because it challenges the view that 5HT act only on the scale of minutes to hours. Our studies motivate the study of rapid 5HT dynamics in earlier paradigms to understand their role in behavioral responses to punishment or patience. Thus, our study reveals avenues to understand the biological roles of 5HT.

Reviewer #2 (Remarks to the Author): In the revised manuscript, with all the changes and the additional analysis provided, all my previous concerns now have been addressed. I have no additional concerns, and I support the publication of the study.

We thank the reviewer for their kind words and insightful suggestions. We are delighted they support publication of our study.

Reviewer #3 (Remarks to the Author):

The authors have adequately addressed all my comments.

We thank the reviewer for their insightful suggestions and kind words for our manuscript.

Reviewer #4 (Remarks to the Author):

The authors have addressed several concerns.

We thank the reviewer for their insightful suggestions and kind words for our manuscript.

Rev4.1. -Given that several reviewers misunderstood the FA/start lick structure it would be useful to better communicate this in the manuscript rather than just in the response to reviewers.

We altered the second paragraph of the results to integrate the difference between these two licks

Licks following pure noise were false alarms (FAs). FAs occurring within the RT window after the presentation of zero-coherence gratings were unrewarded and used to compute psychometric measures (**Fig. 1c**). FAs during the delay period caused trials to restart until the delay period was lick-free (**Fig. 1a, restart**). Most mice FA licked to noise and delay periods were seldom without FAs. We previously showed that many FAs occur because mice react to grating-like features that emerge randomly from the noise. By reverse correlating FAs to the noisy checkerboards, we saw that mice attended to the cued side for increases in verticality and local contrast³⁶. Here, we simplified this previous model to better quantify visual attention.

We have also clarified the label in **Manuscript Fig. 2h** to indicate that it is only the last 15sec of the trial.

References

1. Reynolds, J. H. & Heeger, D. J. The Normalization Model of Attention. *Neuron* **61**, 168–185 (2009).
2. Carandini, M. & Heeger, D. J. Normalization as a canonical neural computation. *Nat Rev Neurosci* **13**, 51–62 (2012).
3. Herrmann, K., Montaser-Kouhsari, L., Carrasco, M. & Heeger, D. J. When size matters: attention affects performance by contrast or response gain. *Nat Neurosci* **13**, 1554–1559 (2010).
4. Denison, R. N., Carrasco, M. & Heeger, D. J. A dynamic normalization model of temporal attention. *Nat Hum Behav* **5**, 1674–1685 (2021).
5. Li, H.-H., Rankin, J., Rinzel, J., Carrasco, M. & Heeger, D. J. Attention model of binocular rivalry. *Proc. Natl. Acad. Sci. U.S.A.* **114**, (2017).
6. Michaiel, A. M., Parker, P. R. L. & Niell, C. M. A Hallucinogenic Serotonin-2A Receptor Agonist Reduces Visual Response Gain and Alters Temporal Dynamics in Mouse V1. *Cell Rep* **26**, 3475-3483.e4 (2019).
7. Barzan, R. *et al.* Gain control of sensory input across polysynaptic circuitries in mouse visual cortex by a single G protein-coupled receptor type (5-HT2A). *Nat Commun* **15**, 8078 (2024).

8. Weber. Htr2a gene and 5-HT_{2A} receptor expression in the cerebral cortex studied using genetically modified mice. *Front. Neurosci.* <https://doi.org/10.3389/fnins.2010.00036> (2010)
doi:10.3389/fnins.2010.00036.
9. Puig, M. V. & Gullledge, A. T. Serotonin and Prefrontal Cortex Function: Neurons, Networks, and Circuits. *Mol Neurobiol* **44**, 449–464 (2011).
10. Eickelbeck, D. *et al.* CaMello-XR enables visualization and optogenetic control of Gq/11 signals and receptor trafficking in GPCR-specific domains. *Commun Biol* **2**, 60 (2019).
11. Celada, P., Puig, M. V. & Artigas, F. Serotonin modulation of cortical neurons and networks. *Front. Integr. Neurosci.* **7**, (2013).
12. Ju, A., Fernandez-Arroyo, B., Wu, Y., Jacky, D. & Beyeler, A. Expression of serotonin 1A and 2A receptors in molecular- and projection-defined neurons of the mouse insular cortex. *Mol Brain* **13**, 99 (2020).
13. Bonnin, A., Peng, W., Hewlett, W. & Levitt, P. Expression mapping of 5-HT₁ serotonin receptor subtypes during fetal and early postnatal mouse forebrain development. *Neuroscience* **141**, 781–794 (2006).
14. Azimi, Z. *et al.* Separable gain control of ongoing and evoked activity in the visual cortex by serotonergic input. *Elife* **9**, e53552 (2020).
15. Hong, S. Z. *et al.* Norepinephrine potentiates and serotonin depresses visual cortical responses by transforming eligibility traces. *Nat Commun* **13**, 3202 (2022).
16. Cohen, J. Y., Amoroso, M. W. & Uchida, N. Serotonergic neurons signal reward and punishment on multiple timescales. *Elife* **4**, (2015).
17. Chebolu, S., Dayan, P. & Lloyd, K. Vigilance, arousal, and acetylcholine: Optimal control of attention in a simple detection task. *PLoS Comput Biol* **18**, e1010642 (2022).
18. Gritton, H. J. *et al.* Cortical cholinergic signaling controls the detection of cues. *Proc Natl Acad Sci U S A* **113**, E1089-1097 (2016).
19. Pinto, L. *et al.* Fast modulation of visual perception by basal forebrain cholinergic neurons. *Nat Neurosci* **16**, 1857–1863 (2013).

20. Sarter, M. & Lustig, C. Cholinergic double duty: cue detection and attentional control. *Curr Opin Psychol* **29**, 102–107 (2019).
21. Parikh, V. & Bangasser, D. A. Cholinergic Signaling Dynamics and Cognitive Control of Attention. *Curr Top Behav Neurosci* **45**, 71–87 (2020).
22. Noudoost, B. & Moore, T. The role of neuromodulators in selective attention. *Trends Cogn Sci* **15**, 585–591 (2011).
23. Ghosh, S. & Maunsell, J. H. R. Locus coeruleus norepinephrine contributes to visual-spatial attention by selectively enhancing perceptual sensitivity. *Neuron* **112**, 2231-2240.e5 (2024).
24. Sara, S. J. & Bouret, S. Orienting and reorienting: the locus coeruleus mediates cognition through arousal. *Neuron* **76**, 130–141 (2012).
25. Aston-Jones, G. & Cohen, J. D. An integrative theory of locus coeruleus-norepinephrine function: adaptive gain and optimal performance. *Annu Rev Neurosci* **28**, 403–450 (2005).
26. Bari, A. *et al.* Differential attentional control mechanisms by two distinct noradrenergic coeruleo-frontal cortical pathways. *Proc. Natl. Acad. Sci. U.S.A.* **117**, 29080–29089 (2020).

Reviewer #1 (Remarks to the Author):

I am very grateful for the time and energy the authors dedicated to addressing my questions. The paper is much clearer in its current version regarding the underlying hypotheses, findings, and conclusions. I have no further questions.

We thank the reviewer for reading our paper and for the insightful suggestions which have improved the clarity and impact of our paper.

In the study “Rapid dynamics of dorsal raphe serotonin neurons selectively regulate the gain of visual attention” by Lehnert and colleagues, the authors show that fast dynamics of serotonin in dorsal raphe and V1 set the attentional gain, independent of location (using calcium imaging in DR, serotonin sensors in V1, and optogenetics). The paper is well written, the results are clear, and the conclusions are very important and well supported by the data. The discussion does a good job at linking these results with the broader literature on the topic.

I only have some minor points to raise, primarily for clarification purposes:

- P2: “Gratings switched sides after ~35 trials and had a flat probability of appearance in time (Fig. 1b).”
Fig 1b does not show that gratings switched nor the flat probability (I think this figure reference should be moved).
- What is the proportion of trials with FAs and misses?
- Is the quantification in Figure 1g in the central bin only? Or average across bins? (Same for Figure 3i and 5k).
- P3: “These results reveal that a second-scale elevation in DR activity precedes a miss, whereas a rapid suppression precedes a hit.”
Why do they show that there is an elevation? With respect to hits? Or with respect to baseline? If the former, then this should be rephrased to reflect what the data shows
- P3: “These results showed that low DR activity predicts higher sensitivity to gratings rather than changes to impulsivity, engagement, or strategy”
Can you clarify which results show which of these claims? Are RT and lapse rate associated with engagement? Response bias to strategy, and criterion to impulsivity?
- P4: “Interestingly, sensitivity to vertical energy and local contrast remained at the center of the screen in both DR high and low (Fig. 3h). This result suggested that rapid DR low dynamics are correlated with increases in attentional gain rather than changes in the way sensitivity was allocated across the screen.”
There are no stats to support this claim. Is the center-bias in DR high significant? (same applies to figure 4h and 4m)
- Can you add some discussion to understand diffused effects of serotonin on the weighted sensitivity maps of 5i-j?